# Association of Intrauterine Microbes with Endometrial Factors in Intrauterine Adhesion Formation and after Medicine Treatment

**DOI:** 10.3390/pathogens11070784

**Published:** 2022-07-10

**Authors:** Ya Wen, Qunfu Wu, Longlong Zhang, Jiangbo He, Yonghong Chen, Xiaoyu Yang, Keqin Zhang, Xuemei Niu, Shenghong Li

**Affiliations:** 1State Key Laboratory for Conservation and Utilization of Bio-Resources, Key Laboratory for Microbial Resources of the Ministry of Education, School of Life Sciences, Yunnan University, Kunming 650091, China; wenya@ydyy.cn (Y.W.); wuqf@mail.ynu.edu.cn (Q.W.); zhanglonglong@ynu.edu.cn (L.Z.); hejiangbo@kmu.edu.cn (J.H.); cyh@ynu.edu.cn (Y.C.); youngman2006@sina.com (X.Y.); kqzhang@ynu.edu.cn (K.Z.); 2Department of Reproduction and Genetics, The First Affiliated Hospital of Kunming Medical University, Kunming 650091, China; 3Kunming Key Laboratory of Respiratory Disease, Kunming University, Kunming 650214, China; 4Regenerative Medicine Research Center, The First People’s Hospital of Yunnan Province, Kunming 650034, China; 5State Key Laboratory of Southwestern Chinese Medicine Resources, Innovative Institute of Chinese Medicine and Pharmacy, Chengdu University of Traditional Chinese Medicine, Chengdu 611137, China

**Keywords:** intrauterine adhesion (IUA), intrauterine microbiota, *Mycoplasmopsis pulmonis* (*Mycoplasma pulmonis*), *Staphylococcus*, *Ulvibacter*, tenascin, estrogen, oxytetracycline, B-cell receptor signaling, estrogen signaling pathway

## Abstract

Intrauterine adhesions (IUAs) have caused serious harm to women’s reproductive health. Although emerging evidence has linked intrauterine microbiome to gynecological diseases, the association of intrauterine microbiome with IUA, remains unknown. We performed metagenome-wide association, metabolomics, and transcriptomics studies on IUA and non-IUA uteri of adult rats to identify IUA-associated microbial species, which affected uterine metabolites and endometrial transcriptions. A rat model was used with one side of the duplex uterus undergoing IUA and the other remaining as a non-IUA control. Both 16S rRNA sequencing and metagenome-wide association analysis revealed that instead of *Mycoplasm**opsis* specie in genital tract, murine lung pathogen *Mycoplasm**opsis*
*pulmonis* markedly increased in IUA samples and displayed a distinct positive interaction with the host immune system. Moreover, most of the IUA-enriched 58 metabolites positively correlate with *M.*
*pulmonis*, which inversely correlates with a mitotic progression inhibitor named 3-hydroxycapric acid. A comparison of metabolic profiles of intrauterine flushing fluids from human patients with IUA, endometritis, and fallopian tube obstruction suggested that rat IUA shared much similarity to human IUA. The endometrial gene *Tenascin-N*, which is responsible for extracellular matrix of wounds, was highly up-regulated, while the key genes encoding parvalbumin, trophectoderm Dkkl1 and telomerase involved in leydig cells, trophectoderm cells, activated T cells and monocytes were dramatically down-regulated in rat IUA endometria. Treatment for rat IUA with estrogen (E2), oxytetracycline (OTC), and a traditional Chinese patent medicine GongXueNing (GXN) did not reduce the incidence of IUA, though inflammatory factor IL-6 was dramatically down-regulated (96–86%) with all three. Instead, in both the E2 and OTC treated groups, IUA became worse with a highly up-regulated B cell receptor signaling pathway, which may be associated with the significantly increased proportions of *Ulvibacter* or *Staphylococcus*. Our results suggest an association between intrauterine microbiota alterations, certain uterine metabolites, characteristic changes in endometrial transcription, and IUA and the possibility to intervene in IUA formation by targeting the causal factors, microbial infection, and Tenascin-like proteins.

## 1. Introduction

Intrauterine adhesion (IUA), also known as Asherman syndrome, is characterized by abnormal endometrial fibrosis joining tissue surfaces of the uterine cavity and is one of the most important causes of female infertility [1,2,3]. IUAs can cause serious harm to women’s reproductive health, including menstrual disorders, low pregnancy rates at 22.5–33.3%, and a high incidence of obstetric complications, such as missed abortion, placenta previa, placental adhesion, premature birth, and postpartum hemorrhage, because IUA hinders the endometrial blood supply and reduces the endometrial receptivity which is not conducive to the growth of the embryos [1,2,3].

In 1927, IUA was reported as a possible long-term complication after surgical termination of pregnancy (TOP) [4,5]. IUAs could occur in 21.2% of women following termination of first trimester pregnancy, and in 12% of women following a pathologically wide internal cervical os. Moreover, 48% of these IUAs were moderate to severe [4,5]. IUA was also detected. Every year about 36–53 million pregnancies have been reported to be terminated worldwide and it has been estimated that 30–50% of women will undergo at least one termination of pregnancy during their lifetime [4,5,6,7]. Because surgical TOP by dilatation and sharp, blunt, or suction curettage has been the effective standard method for TOP since the 1960s, the prevalence of IUA after TOP in women of reproductive age has become a costly public health problem [4,5,6,7].

Iatrogenic endometrial trauma, which could lead to the release of pro-inflammatory and profibrotic cytokines by recruiting and activating the immune system, is assumed to be a key cause of IUAs, because these inflammatory cytokines can result in the accumulation of fibrinogen and extracellular matrix deposition [8,9]. The etiology and progression mechanism of IUAs has been assumed to be similar to that of other damaged tissues and organs with fibrosis [10]. Different from treatment for other organ and tissue fibrosis, the current therapeutic strategy for IUA mainly involves hysteroscopic adhesiolysis combined with postoperative hormone therapies to prevent IUA and to facilitate endometrial regeneration [5,11,12,13,14]. Hormone estrogen has always been used to prevent postoperative re-adhesions because it can promote endometrium growth. However, IUA patients still have a high recurrence rate and poor prognosis and there is controversy due to the opposite effects of estrogen on IUA treatment [15,16,17,18].

Evidence has accumulated that the intrauterine microbiota is an important environmental factor contributing to gynecological diseases, embryo implantation, and pregnancy maintenance [19,20,21]. Previous studies reported that *Mycoplasm**opsis*
*genitalium* was an important cause of pelvic inflammatory disease and acute endometritis [22,23]. Another study suggested that *Ureaplasma urealyticum* infection was one cause of chronic endometritis (CE) since endometritis occurred in 28% of women with *U. urealyticum* present in the chorioamnion at cesarean delivery [24]. Some studies reported that the patients with endometriosis display an increased isolation of *Actinomyces*, *Corynebacterium*, *Enterococcus*, *E. coli*, *Fusobacterium*, *Gardnerella*, *Prevotella*, *Propionibacterium*, *Staphylococcus*, and *Streptococcus* in the endometrium samples and menstrual blood [25]. Antibiogram-driven treatment of chronic endometritis in patients with repeated implantation failure or recurrent pregnancy loss improved their reproductive outcomes [26]. Moreover, patients with CE have a high incidence (46–62%) of moderate and severe IUAs. In particular, the recurrence of IUAs in patients with CE (44.8%) after common interventions for IUA was more than double that of those without CE (20.8%) in a set of prospective cohorts [27]. These strongly suggested that the intrauterine microbiome associated with CE could induce and enhance the occurrence and recurrence of IUAs.

Interestingly, among the microbiota continuum along the female reproductive tract, intrauterine microbiota shared a notable fraction (about 70%) of similar microbiota, including *Pseudomonas*, *Acinetobacter*, *Vagococcus*, and *Sphingobium*, to the microbiota from fallopian tubes and peritoneal fluid from the recto-uterine pouch of Douglas [21], instead of from the vagina/cervical canal. Moreover, the cervical mucus drawn from the cervical canal only contained a very low proportion of *Lactobacillus* which predominated in the vagina. A previous study suggested that IUA patients had a significantly higher percentage of firmicutes in the vagina/cervical canal than non-IUA participants [28], indicating a strong relationship between the microbes along the female reproductive tract and IUA. However, how intrauterine microbiota was involved in IUA formation still remained unknown.

In this study, a rat model was used to avoid interference of individual differences because rats possess a duplex uterus. One side of the duplex uterus underwent IUA model establishment by scratch, and the other side of the duplex uterus in the same animal was used as non-IUA control. We performed 16S rRNA analysis, deep metagenomic sequencing, and metabolomics profiling of the IUA and non-IUA uteri in a cohort of young adult rats. Metabolic analysis of human patients with IUA, endometritis, and obstructed fallopian tubes were also performed to evaluate the similarity of the rat IUA model and human IUA patients. We identified IUA-associated microbial species and their associated effects on IUA formation and transcriptional profiles of host endometria. Our study showed an association between changes in intrauterine microbiota and its metabolic components. Further, we evaluated three medicines, including estrogen (E2), Oxytetracycline (OTC), and a traditional Chinese patent medicine Gongxuening (GXN), on treatment of IUA via IUA diagnostic characteristics combined with 16S rRNA and endometrial transcriptome analysis.

## 2. Materials and Methods

### 2.1. Ethical Approval

This study was approved by the Ethics Committee of the First Affiliated Hospital of Kunming Medical University (NO. (2019)-L-46) and Animal Ethics Committee of Yunnan University (YNUCAE20190005). The sampling was carried out by the Department of Reproductive Genetics, The First Affiliated Hospital of Kunming Medical University and Yunnan University. All patients provided written informed consent.

### 2.2. Study Subjects and Sample Collection

This study was carried out in strict accordance with the recommendations of the Guide for the Care and Use of Laboratory Animals of the National Institutes of Health. The protocol was approved by the Ethics Committee of Yunnan University of Yunnan University. Female adult Sprague Dawley (SD) rats (Shijiazhuang, Hebei, China) weighing 180–220 g, aged 8 weeks, were purchased from Kunming Medical University Animal Laboratory Center (Kunming, Yunnan, China) and were used for the rodent model of IUA [29]. All animals were housed in filter-top cages with autoclaved bedding, autoclaved food and water ad libitum, and a 12 h light/dark cycle. Seventy-two rats were used in the full study. Among them, 49 rats (group one) were used for characterization of the microbial and endometrial factors involved in IUA formation. In 46 rats (group one), the standard scratch method for IUA formation was applied in one uterine horn (total 46 horns) and the other uterine horn without scratch served as a non-IUA control (total 46 horns). The uterine horns from three rats without surgery were used as a control (total 6 horns). Adhesion scores were compared 3 weeks later and 16S rRNA analysis, deep metagenomic sequencing, metabolomics profiling, and endometrial transcriptional analysis of IUA and non-IUA groups were performed. In the second group, 23 animals were operated on. Both uterine horns (total 36 horns) of these 23 rats were inflicted with IUA formation. After 3 weeks, these rats were treated with Estrogen (5 of 23 rats), Oxytetracycline (6 of 23 rats) and Gongxuening (6 of 23 rats), respectively, through oral administration every day for 3 weeks. Six rats treated with water served as a IUA control. Adhesion scores were compared 3 weeks later and 16S rRNA analysis and endometrial transcriptional analysis of the IUA and non-IUA groups were performed.

The standard scratch method for IUA formation was carried out according to a previous study [29]. After administration of ketamine/xylazine by intraperitoneal injection, a vertical incision of 3 cm was made in the abdominal wall to expose the duplex uterus. A small incision of 0.5 cm was made in one of the uterine horns at the utero-tubal junction and a 27 Gauge needle was inserted two-thirds of the way through the uterine cavity in a standardized fashion for traumatizing one uterine horn of the duplex uterus, by rotating and pulling the needle four times. The other uterine horn was not touched and served as the control. Following four estrous cycles after the operation on uteri, the uterine horns were collected. Then, the endometrium samples were collected with a sterile mini-endometrial curette via scrapping off from inter-uterine horns [29]. All the samples were put in 2 mL cryogenic vials (Corning, New York, NY, USA) and stored at −80 °C until analysis. All the endometrium samples were histologically evaluated for the diagnostic characteristics for IUA.

The human samples and clinical information used in this study were obtained under conditions of informed consent, and with approval of the human ethical committee of The First Affiliated Hospital of Kunming Medical University. Subjects undergoing total hysteroscopy in the First Affiliated Hospital of Kunming Medical University from February 2020 to April 2020. Prior to total hysteroscopy, patients first were checked by two clinicians and their diagnosis was further confirmed by pathology reports. The patients with endometritis were further confirmed by two histopathologists. Healthy women who underwent testing for tubal patency and hysteroscopy served as the control group. All the IUA women aged ≤40 years with no hormonal treatment for at least 3 months before surgery were included in the study (Appendix A). Women displayed regular menstrual cycles without steroid treatment or other medication for at least 3 months before the collection of tissue. Patients who had a combination of other uterine diseases or two or three of these conditions, patients with polycystic ovary and ovarian tumors, or patients with metabolic diseases (such as thyroid disease) were excluded in this study.

On the day of hysteroscopy, uterine flushing fluid from human patients were collected according to a previously described protocol [30]. Briefly, a sterile speculum was placed in the vagina, visualizing the cervical os and positioning an insemination catheter into the uterine lumen. The catheter was connected to a 10 mL syringe filled with 3.5 mL of sterile normal saline solution. The saline solution was slowly infused into the uterine cavity and aspirated. The procedure was repeated five times to achieve turbulent flow and homogeneous distribution of sample within the fluid. The fluid was then collected, centrifuged at 3× *g* for 3 min, and stored in a 1.5 mL microfuge tube (Thermo Fisher Scientific, Waltham, MA, USA) at −80 °C until analysis.

### 2.3. Histological and Immunohistochemistry Analysis of Rat IUA Endometria

Endometrial tissue was dehydrated and embedded in paraffin, then sectioned and examined histologically by hematoxylin and eosin (H&E); morphological changes were examined using a microscope (Olympus BX41 microscope, Olympus Corporation, Japan); and images were captured by a digital camera (Fuij Co, Sapporo, Japan).

For immunohistochemistry, the paraffin-embedded endometrial sections were deparaffinized, rehydrated, and immersed in H_2_O_2_ (3%) for 10 min, to block any peroxidase activity. Slides were washed in phosphate buffer saline (PBS). Nonspecific binding sites were blocked by bovine serum albumin (5%) prior to the addition of TGF-β1 polyclonal antibody in a dilution of 1:300 overnight at 4 °C. The slides were then washed in PBS and incubated with a secondary antibody, developed with 3,3′-diaminobezidine tetrahydrochloride and counterstained with hematoxyline. The immunopositive cells in the endometrium were counted in 10 high-power fields (x 400) by independent researchers blinded to information pertaining to the study, and the percentage of immunopositive cells was calculated by dividing the number of immunopositive cells by the total number of cells and then multiplying the resulting number by 100.

### 2.4. DNA Extraction and 16S rRNA Sequencing of Rat IUA Endometria

Microbiome characterization was performed by Biotree Inc. (Shanghai, China). Rodent specimens were placed into a MoBio PowerMag Soil DNA Isolation Bead Plate. DNA was extracted following MoBio’s instructions on a KingFisher robot. The V4 hypervariable region (515–806 nt) of the 16S rRNA gene was amplified using universal bacterial primers pair (515F: 5′-GTGCCAGCMGCCGCGGTAA. 806R: 5′-GGACTACHVHHHTWTCTAAT). Amplicons were sequenced at Illumina MiSeq platform using the 250-bp paired-end kit (v.2). Sequences were denoised, clustered into 97%-similarity operational taxonomic units (OTUs) with the Mothur software package (https://www.mothur.org/wiki/MiSeq_SOP; accessed on 12 November 2018) (v. 1.39.5), and taxonomically classified using Greengenes (v.13.8) as the reference database. The possibility for contamination examined by co-sequencing DNA amplified from samples and from four each of template-free controls and extraction kit reagents treated the same way as the samples. Among them, the abundance measures in a single sample by estimating the outnumber, and the diversity index measures the heterogeneity of communities.

### 2.5. Shotgun Metagenomic Sequence Data Analysis

The raw paired-end Illumina obtained from whole shotgun metagenome sequencing of each metagenomic sample were preprocessed using fastp (v.0.23.2) [31] to remove the sequencing adaptor and low-quality sequences. The filtered reads were aligned against the *Rattus norvegicus* complete genome (NCBI Project Accession Number: PRJNA677964) to eliminate contaminated sequences deriving from host genome DNA. Taxonomic classification from metagenomics reads was performed using Kraken2 (v. 2.1.2) [32], a k-mer-based sequence classification approach, against standard RefSeq databases (release 20210517). For Bracken abundance estimation, we generated a Bracken-database file using bracken-build on the above Kraken database with a k-mer length of 50, 75, 100, 150, 200, 250, and 300 bp and estimate taxonomic abundance down to the species level. Microbial community profiles at species level were compared using relative abundance. The Bray–Curtis distance between samples from Con, Non, and IUA rat was computed using beta_diversity.py script on QIIME1.9.1, and principal coordinate analysis (PCoA) was used to visualize the microbial community hierarchies. Alpha diversity indices were computed using alpha_diversity.py scripts. LefSe analysis was conducted to find microbial species that consistently represent the differences among communities. All the plots and relevant statistical analysis were performed on R software (v4.1.2).

### 2.6. Construction of Phylum, Genus, and Species and KEGG Orthology (KO) Profiles

The high-quality reads were aligned to the updated Genital tract microbiome gene catalog by SOAP2 with the criterion of more than 95% identity and 90% overlap of query. Sequence-based gene abundance profiling was performed as previously described [30,33], and there was a total of 213,414 genes in 20 samples. The relative abundances of phyla, genera, species, and KOs were calculated from the relative abundance of their respective genes using previously published methods. Gene count and α-diversity were analyzed as previously described [30,33].

### 2.7. Functional Profiling

Open reading frames (ORFs) were predicted on the obtained scaffolds using MetaGeneMark (version 3.26). The ORFs were compared with those of Bacteria, Fungi, Archaea, and Viruses extracted from Non-Redundant Protein Sequence Database Version: 2018.0, database of NCBI using DIAMOND (version 0.9.10) (blastp, E < 1 × 10^−5^). Gene abundances were computed as follows: High-quality reads were mapped back onto the scaffolds using Bowtie 2 version 2.2.9. Each ORF on each scaffold was scored for read coverage, which was defined as the number of base pairs mapped onto the corresponding scaffold regions divided by the lengths of the ORFs. When more than two ORFs match up to one gene, each gene abundance was computed as the average of the score values. All genes in our catalogue were translated to amino acid sequences and aligned to the KEGG database version 59 using USEARCH10 (E < 1 × 10^−5^). Each predicted protein was assigned a KO based on the best hit gene in the KEGG database. The abundance of a KO was calculated by summing the abundance of genes annotated to a feature.

### 2.8. Metabolomics Profiling with UHPLC-QTOF-MS

Intrauterine tissue samples (100 mg) from rodents were individually ground with liquid nitrogen and the homogenate was re-suspended with pre-chilled 80% methanol and 0.1% formic acid by well vortexing. The samples were incubated on ice for 5 min and then were centrifuged at 15,000 rpm and 4 °C for 5 min. An aliquot of supernatant was diluted to the final concentration containing 53% methanol by UHPLC-MS grade water. The samples were subsequently transferred to a fresh Eppendorf tube and then were centrifuged at 15,000× *g*, 4 °C for 10 min. Finally, metabolic analysis of the supernatant was conducted using the LC-QTOF-MS system (Palo Alto, CA, USA).

### 2.9. Metabolomics Data Analysis

Peaks were detected after relative standard deviation de-noising. Then, the missing values were filled up by half of the minimum value. Additionally, the internal standard normalization method was employed in this data analysis. The final dataset containing the information of peak number, sample name, and normalized peak area was imported to SIMCA15.0.2 software package (Sartorius Stedim Data Analytics AB, Umea, Sweden) for multivariate analysis. Data were scaled and logarithmically transformed to minimize the impact of both noise and high variance of the variables. The resulting metabolites were annotated using the KEGG database (http://www.genome.jp/kegg/; accessed on 15 April 2019), HMDB database (http://www.hmdb.ca/; accessed on 15 April 2019), and Lipidmaps database (http://www.lipidmaps.org/; accessed on 15 April 2019). After these transformations, principal component analysis (PCA), an unsupervised analysis that reduces the dimension of the data, was carried out to visualize the distribution and the grouping of the samples. A 95% confidence interval in the PCA score plot was used as the threshold to identify potential outliers in the dataset. Analysis of PCA and orthogonal projections to latent structures-discriminate analysis (OPLS-DA) with SIMCA software (v15.0.2, Umea, Sweden) was used to perform log conversion and CTR formatting on data. PCA score graph displayed that both groups of samples were in the 95% confidence interval (Hotelling’s t-squared ellipse).

Supervised PLS-DA was utilized to analyze the group separation of metabolomics data and mine the variables responsible for classification. The robustness and predictive ability of the estimation model were verified by seven-fold cross-validation, and the model was further validated by permutation test with 200 permutations. Afterward, the R2 and Q2 intercept values were obtained. Here, the intercept value of Q2 represents the robustness of the model, the risk of overfitting, and the reliability of the model, which will be better if smaller. R2Y is 0.73, which indicates that the model can better explain the difference between the two groups of samples. The intercept between the regression line and the vertical axis of Q2 is less than zero (−0.74). Meanwhile, with the decrease in replacement retention, the proportion of the y-variable increases, and the Q2 of the random model decreases, indicating that the model has good robustness and there is no over fitting phenomenon.

Using the first principal component of Supervised OPLS-DA model, the value of variable importance in the project (VIP) is used. The VIP value represents the contribution rate of metabolite difference in different groups; the multiple of difference (fold change, FC) is the ratio of the mean values of all biological repeated quantitative values of each metabolite in the comparison group, and the *p* value of student *t*-test is used to find the differential expression metabolites. The threshold value is set to VIP > 1.0, and *p* < 0.05. Metabolites of interest which based on Log2 Fold Change and −log10 (*p* value) of metabolites were filtered. For clustering heat maps, the data were normalized using z-scores of the intensity areas of differential metabolites and were plotted by Pheatmap package in R language. Commercial databases, including KEGG (http://www.genome.jp/kegg/; accessed on 18 April 2019) and Metabo-Analyst (http://www.metaboanalyst.ca/; accessed on 18 April 2019), were used for pathway enrichment analysis.

### 2.10. Associations between Microbiome and Metabolites

Spearman’s rank correlation coefficient was used to interpret the correlation between the microbial abundance profiles of the samples and the detected endometrial metabolites. Clinical indices with *p* < 0.05 and the absolute value of Spearman’s rank correlation coefficient (R) > 0.2 were considered to associate with respective microbial abundance profiles.

### 2.11. Transcriptomics

A Trizol-based method was used to extract the total RNA from rodent endometria. The ratio of the absorbance at 260 and 280 nm (A260/280) ranged from 1.91 to 2.03 (as determined by NanoDrop ND-1000 spectrophotometer; NanoDrop Technologies LLC). Each library was quantified by fluorimetry (Qubit quant-iT HS dsDNA reagent kit, Invitrogen). RNA integrity number (RIN) ranged from 6.8 to 9.2, and 28S:18S ratios ranged from 0.6 to 1.7 (as determined by the Fragment Analyzer instrument; Advanced Analytical Technologies, Inc., Ames, IA, USA). Sixteen samples were submitted for RNA library preparation using Illumina’s TruSeq mRNA stranded sample preparation kit at the University of Missouri DNA Core Facility. The libraries were sequenced using an Illumina NextSeq 500 sequencer to generate >45 million 75-bp single-end reads per sample. The raw sequences (FASTQ) were subjected to a FastQC (www.bioinformatics.babraham.ac.uk/projects/fastqc/; accessed on 11 January 2019) tool for checking sequence quality. The adapter sequences were removed by Cutadapt. The program Fqtrim (https://ccb.jhu.edu/software/fqtrim/; accessed on 11 January 2019) was used to perform quality trimming (Phred score >30) by a sliding window scan (6 nucleotides), and to remove reads shorter than 20 bp. Reads obtained from the quality control step were mapped to *Rattus norvegicus* complete genome (NCBI Project Accession Number: PRJNA677964) using Hisat2 aligner. The gene annotation along with the alignment files were used in FeatureCounts tool to quantify reads that mapped to each gene of each sample.

### 2.12. Medicine Treatments for IUA

After we found an obvious variety in intrauterine microbes in rat IUA models, another 23 rat models with IUA were successfully established according to the above method and were used for treatment of the dominant microbe in IUA. The 23 IUA rats were randomly divided into four groups. The rats in the E2 group were fed with oral estrogen (Progynova, Bayer, Germany) at 0.01 mg/kg (0.003 mg per day per rat) for three estrous cycles, rats in the OTC group were fed with oxytetracycin (PingGuang Pharmaceutical, Xuzhou, China) at a dose of 15 mg/kg (3 mg/d) for three estrous cycles, and rat GXN group were fed with Chinese medicine GongXueNing (Yunnan Baiyao, Kunming, China) at 65 mg/kg (13 mg/d) for three estrous cycles. IUA Rats in the control group were fed with sterile distilled water at a dose of 2 mL/d for three estrous cycles. The endometrial tissues were taken for further analysis after three estrous cycles (about 21 days).

### 2.13. Statistical Analysis

When comparing the differences of a component between groups, mean was used to represent the value of that in the group, and standard deviation (SD) was shown in the figure as error bars. Microbial community differences analysis between groups were computed using permutational multivariate analysis of variance (PERMANOVA). Statistical analyses were performed using GraphPad Prism version 8.0 software or R packages. Benjamini-Hochberg (BH) method was used to adjust *p* value computed from corresponding significant differences test. The details of the tests used are included in the figure legends.

## 3. Results and Discussion

### 3.1. IUA in the Rat Model Was Successfully Constructed with Patient-Matched Characteristics

One horn of the uterus of 46 rats of 8 weeks with an average weight of 200 g was established with IUA and the other horn without surgical operation was used as control. The endometrium samples in the uteri of all the rats were collected according to the previous method [29]. The myometria and endometria in the non-IUA control group were clearly demarcated, and endometrial stromal cells were orderly distributed with normal size (Figure 1A–D). The glandular and endometrial epithelia were intact, and no obvious fibrosis was observed in the non-IUA control group. Moreover, endometrial glands in the IUA group decreased at 10.16 ± 8.56 with 55% lower than those in the non-IUA group at 4.58 ± 4.56 (Figure 1E). Micro-vessels (19.4 ± 5.78) in the IUA group were 24% higher than those in the non-IUA control group (24.43 ± 6.89) (Figure 1F). Importantly, the fibrotic area in the IUA group (38.71 ± 12.6) was 134.6% higher than that in the non-IUA control group (16.5 ± 7.75%) (Figure 1G). The immunoreactivity level for TGF-β1 expression in endometrial epithelium of the IUA group was about 40% while no immunoreactivity for TGF-β1 expression in the non-IUA group was observed (Figure 1H). These results suggest that IUA in the rat model was successfully constructed, and matched diagnostic characteristics observed in human patients [13].

### 3.2. Rat IUA-Associated Microbes Identified by 16S rRNA and Metagenome-Wide Association Analysis

The 16S rRNA analysis revealed 5549 operational taxonomic units (OTUs) found in both groups, of which 804 OTUs were found only in IUA and 803 OTUs were found only in non-IUA (*p* < 0.05). Comparison of the microbial community composition in the IUA and non-IUA groups displayed a significant difference between these two groups (Appendix A).

At the phylum level, Proteobacteria (25.82% in IUA vs. 27.29% in non-IUA), Firmicutes (21.93% in IUA vs. 19.54% in non-IUA), Acidobacteria (12.75% in IUA vs. 14.07% in non-IUA), Bacteroidetes (10.63% in IUA vs. 10.45% in non-IUA) Chloroflexi (4.02% in IUA vs. 4.22% in non-IUA), Actinobacteria (3.42% in IUA vs. 3.96% in non-IUA), Tenericutes (3.31% in IUA vs. 1.82% in non-IUA), Zixibacteria (2.31% in IUA vs. 2.58% in non-IUA), and Spirochaetes (1.30% in IUA vs. 1.31% in non-IUA) constituted nine of the most predominant phyla in both groups (Appendix A). Among them, IUA samples contained significantly increased proportions of Firmicutes and Tenericutes, with 12.23% and 81.87%, respectively, higher levels than non-IUA samples, indicating that Tenericute was the greatest IUA-enriched phylum.

At the genus level, *Mycoplasmopsis* (2.93% in IUA vs. 1.48% in non-IUA), *Occallatibacter* (2.41% in IUA vs. 2.60% in non-IUA), unidentified *Ruminococcaceae* (2.13% in IUA vs. 2.00% in non-IUA), *Silvanigrella* (1.78% in IUA vs. 1.94% in non-IUA), unidentified *Lachnospiraceae* (1.70% in IUA vs. 1.40% in non-IUA), *Faecalibacterium* (1.09% in IUA vs. 0.93% in non-IUA, *Agathobacter* (0.80% in IUA vs. 0.81% in non-IUA), *Lactobacillus* (0.71% in IUA vs. 0.65% in non-IUA), and *Bifidobacterium* (0.48% in IUA vs. 0.57% in non-IUA), constituted nine of the most predominant genera in IUA and non-IUA groups (Appendix A). The proportions of *Mycoplasmopsis* (*Mycoplasma*) in IUA samples were almost twice as high as that in non-IUA ones.

Shotgun sequencing analysis of the microbiomes of the IUA and non-IUA groups were also performed, and the uterine horns from rats without surgery were used as a pre-control (Con). PCoA plot based on Bray–Curtis distances displayed a clear separation of the Con group from IUA and non-IUA groups, indicating that the microbial community in the Con group was distinct from those in the IUA and non-IUA groups (Figure 2A). Additionally, there was a significant difference in the microbial community between the IUA and non-IUA group. Moreover, the IUA group had lower bacterial richness and diversity than non-IUA and Con groups (Figure 2B,C; Appendix A).

Consistent with 16S rRNA analysis, Tenericute was the phylum with the greatest change in proportion from 1.866% ranking second in the non-IUA controls to 5.617% ranking first in the IUA (Appendix A) (*p* < 0.05). *Mycoplasmopsis pulmonis* (*Mycoplasma pulmonis*) was the predominant species and dominated the increased proportion of the genus *Mycoplasmopsis* in IUA samples (Figure 2D and Appendix A). The relative abundance of *M. pulmonis* in the IUA group was 5.59%, three times the 1.86% in the non-IUA control group. Comparison of the Pearson correlation of the top four dominant species with microbial diversity suggests that only *M. pulmonis* displays a strong negative association with microbial diversity (Figure 2E), indicating the state of microbial dysbiosis in IUA samples was induced by *M. pulmonis*.

### 3.3. Functional Characterization of the Rat IUA-Associated Microbial Genes

A total of 495 microbial genes present in 92 samples were identified in IUA and non-IUA samples and 19 genes were most differentially enriched between IUA subjects and non-IUA controls with metastats analysis (*p* < 0.05; Figure 3A; Appendix A). Highly enriched genes in IUA vs. non-IUA included K02057 (assigned as F-type H^+^/Na^+^-transporting ATPase subunit alpha) involved in the phosphotransferase system pathway; K07497 (putative transposase) involved in nucleotide excision repair; K00382 (dihydrolipoamide dehydrogenase) involved in pyruvate metabolism; and K03475 (ascorbate PTS system EIIC component) involved in the phosphotransferase system (Figure 3A). Two of these four IUA-enriched genes, K00382 and K03475, were ascribed to *M. pulmonis* (Figure 3A; Appendix A). All the 15 control-enriched genes were not found to be associated with *M. pulmonis* with the most abundant gene, K04961 (assigned as Ryanodine receptor 1) being positively associated with the control-enriched *Escherichia coli* (4.69‱ in non-IUA vs. 4.46‱ in IUA. One other control-enriched gene, K00933 assigned as creatine kinase was ascribed to the control-enriched *Proteobacteria bacterium* (1.15‱ in non-IUA vs. 0.77‱ in IUA) (Appendix A).

Comparison analysis displayed that the top ten abundant genes in both groups were all significantly highly enriched in the IUA group compared to the non-IUA control group (Figure 3B; Appendix A). Among them, seven genes were also ascribed to *M. pulmonis*, including K10112 (assigned as a multiple sugar transport system of ATP-binding protein) and K15583 (assigned as leucyl aminopeptidase), both involved in ABC transporters and Quorum sensing [34]; K03657 (assigned as putative transposase) in nucleotide excision repair pathway; K03763 (assigned as DNA polymerase III subunit alpha, Gram-positive type) in DNA replication pathway; K03046 (assigned as DNA-directed RNA polymerase); and K03427 (assigned as F-type I restriction enzyme M protein) both in purine metabolism and pyrimidine metabolism; and K02111 (assigned to ATPF1A, F-type H^+^/Na^+^-transporting ATPase subunit alpha, atpA) in phosphotransferase system (Figure 3C,D; Appendix A). In all, *M. pulmonis* in IUA endometria was positively correlated with half of highly increased microbial genes, including 70% of the top ten abundant genes in IUA microbiota. These highly enriched genes were responsible for the energy and supply required for growth and proliferation of *M. pulmonis* in host [35].

### 3.4. Associations of Rat IUA Microbial Species with Uterine Metabolites

Non-targeted metabolomics profiling of the IUA and controls were performed. Using mass spectrometry with positive ion mode (POS) and negative ion mode (NEG), we identified 153 and 199 metabolites, respectively, that significantly differed in abundance between the IUA and non-IUA control (VIP > 1.0 and *p* < 0.05). Among these metabolites, 35 compounds were structurally identified with the coupled tandem mass spectrometry with POS (Figure 4A; Appendix A) and 29 with NEG (Figure 4B; Appendix A). Most of the identified metabolites were significantly increased, while only a few were decreased, in the IUA vs. non-IUA control. All the significantly enriched metabolites in IUAs were located in two KEGG pathways, biosynthesis of fatty acid and unsaturated fatty acids (Figure 4C,D; Appendix A).

Most of the 58 metabolites found enriched in IUA-enriched samples were positively associated with *M. pulmonic*, and six IUA-diminished metabolites were inversely correlated with *M. pulmonis*. Among them, 11 IUA-enriched metabolites exhibited significant association with *M. pulmonis* (Spearman’s correlation, * *p* < 0.05, Figure 4E,F; Appendix A); including His-Gln, His-Thr; His-Ser, Sphingosine; Arachidonoylglycine; 2′-deoxy-D-ribose; 25-hydroxyvitamin D3; Vitamin E; all *cis*-(6,9,12)-Linolenic acid; D-xylulose, and 3-hydroxydodecanoic acid; and one IUA-diminished metabolite, 3-hydroxycapric acid, a known inhibitor for mitotic progression [36]. 

Close positive associations of the non-IUA control-enriched metabolites with the five non-IUA control-enriched microbiome species, including *Oceanospirillum multiglobuliferum* (Pentadecanoic acid); *Chlamydia abortus* (Pentadecanoic acid); *Cyberlindnera jadinii* (Urocanic acid); *Piromyces finnis* (Phosphocreatine, Arg-Ser, His-Gln, His-Thr, N-methylhydantoin, L-arginine, methylmalonic acid); and *Mircobactium profundi* (Acetylcarnitine, D-mannose, γ-Tocotrienol, Vitamin E), were also observed (Spearman’s correlation, *p* < 0.05; Figure 4E,F). However, these species were scarce, especially the last three, in both the IUA and non-IUA control groups (*Chlamydia abortus* 0.90% in IUA vs. 0.97% in non-IUA control); (*Oceanospirillum multiglobuliferum* 0.14% in IUA and 0.15% in non-IUA control; *Piromyces finnis* 0.0038% in IUA vs. 0.0079% in non-IUA control; *Cyberlindnera jadinii* 0.008% in IUA vs. 0.01% in non-IUA control; and *Microbacterium profundi* 0.007% in IUA vs. 0.01% in non-IUA control) (Figure 2).

In all, the IUA-enriched *M. pulmonis* displayed a significantly positive association with about one fifth of the IUA-enriched 58 metabolites and a significantly negative association with one sixth of the IUA down-regulated six metabolites in the uteri. The IUA highly enriched metabolites significantly associated with *M. pulmonis* included the unique lipoproteins or lipopeptides involved in *Mycoplasmopsis* membrane. Importantly, the only one IUA which down-regulated metabolite 3-hydroxycapric acid, an inhibitor for mitotic progression, was also significantly associated with *M. pulmonis*.

### 3.5. Similar Metabolites between Human IUA and Rat IUA while Distinct Metabolic Profiles between Human IUA and Human Endometritis

Previous studies reported that *M. genitalium* plays an important role in pelvic inflammatory disease and acute endometritis [22,23]. We wondered if there were any differences between the IUA-associated *Mycoplasmopsis* species and the endometritis-associated *Mycoplasmopsis* species in humans. To collect any endometrium samples from these patients’ uteri with any surgical tools was infeasible due to potential trauma injury. Only medical flushing fluids that could be allowed to flush patient’s endometria were accessible. Thus, we collected the intrauterine flushing fluids from human patients with IUAs (n = 5), endometritis (EM, n = 8), and fallopian tube obstruction (FTO, n = 4), and compared metabolic profiles of these samples. A total of 307 metabolites significantly differed in abundance in human IUA vs. FTO; 235 metabolites in human IUA vs. EM; and 93 metabolites in EM vs. FTO. Among these metabolites, 90, 42, and 10 compounds, respectively, were structurally identified with the coupled tandem mass spectrometry with POS (VIP > 1.0 and *p* < 0.05) (Appendix A). Obviously, the number (ten) of the differentially regulated metabolites in EM vs. FTO were far less than those of human IUA vs. FTO or EM, indicating that IUAs in humans could cause much more metabolic alteration than endometritis and FTO. Among them, ten significantly up-regulated metabolites in human IUA patients, including Ser-Arg, beta-citronellol, D-mannose, Ile-Asn, Acetylcarnitine, Urea, Met-Gln, creatinine, uridine diphosphate, and phosphocreatine were also found in the above rat IUAs. Moreover, 89 out of the 90 metabolites and 32 of the 42 compounds were found significantly increased in human IUA compared to FTO and endometritis, respectively. The similar IUA-associated metabolites between humans and rats suggests the possible role of *Mycoplasmopsis* involvement in human IUAs. The distinct metabolic alteration between human IUA and human endometritis indicated that the endometritis-associated *M**. genitalium* in humans might not be associated with IUA.

### 3.6. The Association of IUA with Rat Endometrial Transcriptome

Transcriptome analysis of IUA and non-IUA control endometria displayed that 504 differentially expressed genes (DEGs) were significantly up-regulated and 186 DEGs were significantly down-regulated in IUA endometrial (DESeq, *p* ≤ 0.05; Figure 5; Appendix A). Among the top 20 IUA-enriched DEGs (Figure 5A; Appendix A), except one unknown DEG, 12 DEGs were assigned as Ig kappa chain V19-17 like (LOC100912707): ATPase/H^+^ transporting lysosomal V0 subunit D2 (Atp6v0d2, restricting inflammasome activation and bacterial infection by facilitating autophagosome-lysosome fusion); secreted frizzled-related protein 5 (Sfrp5); signaling lymphocyte activation molecule family member 6 (Slamf6); Cd5 molecule (Cd5l) similar to RIKEN cDNA (MGC94199); chemokine C-C motif ligand 9 (Ccl9); guanylate binding protein 1/interferon-inducible (Gbp1); immunoglobulin joining chain/Uncharacterized protein (Igj) similar to PIRB1 (RGD1566307); immunoglobulin kappa constant (Igkc); and chemokine C-X-C motif ligand 9 (Cxcl9), respectively, which were all involved in immune responses. Importantly, one DEG assigned to tenascin N, belonging to a family of four giant proteins with six arms allowing cell rounding and promoting condensation [37,38,39], was among the top ten IUA-enriched DEGs.

Among the top 20 IUA-diminished DEGs in endometria (Figure 5B; Appendix A), except for eight unknown DEGs, five DEGs were assigned to parvalbumin (Pvalb): calcium channel/voltage-dependent L type alpha 1S subunit (Cacbals); Dickkopf-like 1 (Dkkl1), telomerase (Terc); and testis-specific serine kinase 5 (Tssk5), respectively. These DEGs were mainly involved in the production of Leydig cells and trophectoderm cells, and in the immortalization of activated T cells and monocytes. Parvalbumin played an important role in the control of the excitability and activity of GABAergic neurons and the number of the Leydig cell population, Ca^2+^-dependent Cacbals testosterone secretion and LH receptor content increased with parvalbumin levels [40]. Dickkopf-like 1 (DkkL1) was reportedly found to be involved in two seemingly unrelated functions: the production of sperm and the production of trophectoderm cells and their derivatives [41]. Importantly, DkkL1 appeared in the trophectoderm and eventually in the trophoblast giant cells that are involved in implantation [41]. Telomerase (Terc) is active only in germinal tissues, stem cells and their immediate progeny, activated T cells, and monocytes [42]. In somatic cells, telomerase are repressed and immortalized cell lines and around 85–90% of cancer cells maintain their telomeres through the activation of telomerase (Gene Therapy Targeting Receptor-Mediated Cell Death to Cancers). Testis-specific serine kinase 5 (Tssk5) was involved in spermatogenesis through the phosphorylated CREB pathway [43]. Here, the above DEGs that were mainly involved in the production of Leydig cells and trophectoderm cells and in the immortalization of activated T cells and monocytes, respectively, were all significantly down-regulated in the IUA group.

Moreover, the down-regulated DEGs in IUA endometria also included mitochondrial creatine kinase (Ckmt2) and spermidine synthase-like (LOC100912604), both for protection against hypoxia/reoxygenation injury, NADH dehydrogenase (ubiquinone) 1 alpha subcomplex (Ndufal) in mitochondria, and the member 2 of nuclear receptor subfamily 1 group I (Nrli2), encoding a prominent xenosensor regulating the expression of biotransformation enzymes in detoxification, such as glutathione S-transferase Mu 5 (GSTM5) (Appendix A). All the above six genes were mainly involved in the detoxification process.

To interpret the biological impact of the DEGs in IUA vs. non-IUA control, Gene Set Enrichment Analysis (GSEA) was performed to retrieve the functional profile associated with the DEGs. Thirty-three pathways were up-regulated and only one pathway ribosome (RNO 03010) was down-regulated in IUA vs. non-IUA (Figure 5C,D; Appendix A). The highly IUA-enriched pathways included cytokine–cytokine receptor interaction, chemokine signaling pathway, *Staphylococcus aureus* infection, hematopoietic cell lineage, natural killer cell mediated cytotoxicity, T cell receptor signaling pathway, complement and coagulation cascade, B cell receptor signaling pathway, Th17 cell differentiation, Th1 cell differentiation, and Th2 cell differentiation. Among them, most of the IUA-enriched pathways were mainly involved in immune systems. Cytokine–cytokine receptor interaction was found to be involved in tacrolimus-induced renal fibrosis progression [44], and in inflammation and severe fibrosis in nonalcoholic steatohepatitis [45]. The increase in *Staphylococcus*
*aureus* infection was also consistent with a previous study that observed *Staphylococcus*
*aureus* could induce acute exacerbation of pulmonary fibrosis [46].

A previous study reported that the inflammatory factor NF-κB expression was significantly elevated in the IUA endometrial [47]. Here in this study, we also found that the inflammatory factor NF-κB expression was among those significantly elevated in the IUA endometrium samples, but with a ranking of 27. Thus, NF-κB expression was not included in Figure 5A.

### 3.7. Effects of E2, OTC and GXN on IUA Diagnostic Characteristics

Estrogen (E2) therapy has been widely used for IUA treatment [11,12,13,14]. Oxytetracycline (OTC) has been applied for *Mycoplasmopsis* infection, in particular, in cases where macrolide resistance was suspected [48,49]. A previous study reported that *Mycoplasmopsis* infection is associated with reproductive failure in a number of mammals, and the administration of OTC to infertile couples could result in about a 30% pregnancy increase [50]. Gongxuening (GXN), a traditional Chinese patent medicine (TCPM), are commonly used to treat pelvic hemostasis and inflammation [51]. Thus, we examined the effects of these three medicines on IUA treatment after the IUA rats were given these three medicines by gavage for 21 days (about four estrous cycles), and the control group was given sterile distilled water. The uterine tissues were collected for IUA diagnostic characteristics, 16S RNA, and endometrial transcriptome analysis.

In contrast to our expectation, both E2 (n = 5) and OTC (n = 6) groups displayed dramatically decreased endometrial glands with 40% and 20% lower than the IUA control (n = 6) (Figure 6A), respectively. Meanwhile, the GXN group (n = 6) showed endometrial glands similar to the IUA control (Figure 6A). Importantly, the IUA areas in E2 (53%) and OTC (72%) groups were 120.8% and 200.0% higher than that in the IUA control (24%) (Figure 6B, Student’s *t*-test. *p* < 0.05, *p* < 0.01, n = 5). At the same time, TGF-β1 in both the E2 and OTC groups were highly increased with 72.5% and 91.5% higher than that in the IUA control group (Figure 6C). These results suggested both E2 and OTC treatments could not decrease IUA and increase endometrial glands.

### 3.8. Effects of E2, OTC and GXN on the IUA Microbiota

A 16S RNA analysis revealed that there were 3504 OTUs with 1264 only found in the IUA control group, 2341 with 577 in the E2 group, 2723 with 764 in the OTC group, and 3373 with 1056 in GXN group (Figure 7A). The species abundances in all the three medicine treated groups were significantly lower than that in IUA control group (Figure 7B; Appendix A). The species diversity in E2 group was a little higher than that in the IUA control group while those in the OTC and GXN groups were a litter lower than that in the IUA control group (Figure 2C; Appendix A). The distribution and composition of species in the IUA control group and GXN groups seemed much larger than those in the other two medicine-treated groups, E2 and OTC (Figure 7D,E), indicating a significant difference among these four groups.

At the phylum level, consistent with the above IUA results, Proteobacteria and Firmicutes were the first two dominant phyla in all the four groups. However, Proteobacteria was the first predominant phylum in three groups with 39.67% in the IUA control, 41.99% in OTC and 41.14% in GXN, respectively, and the second predominant phylum at 27.73% in E2 (Figure 7F; Appendix A). Firmicutes was the first predominant phylum in E2 at 30.78% and the second predominant phylum in the other three groups with 19.37% in the IUA control, 25.14% in OTC and 21.74% in GXN, respectively (Figure 7F, Appendix A). These results suggest that E2 can largely decrease Proteobacteria microorganisms by 30.1%. and increase Firmicutes microorganisms by 58.9%, while GXN has little effect on species compositions.

At the genus level, notably, *Mycoplasmopsis* was no longer the dominant genus in all the four groups after treatment with medicines or sterile distilled water. Previous studies reported that *Mycoplasmopsis* infection is usually resolved spontaneously due to the self-resolving nature that *Mycoplasmopsis* would die on their own normally within 3 weeks [52,53], consistent with our result. *Ralstonia*, *Pseudogracilibacillus*, and *Corynebacterium* were the most dominant three genera in all the four groups after treatments, in contrast to the fact that *Mycoplasmopsis*, *Occallatibacter*, and the unidentified *Ruminococcaceae* were the top three dominant genera in IUA groups before treatment. The first predominant genus *Ralstonia* microorganisms in the E2 group decreased by 31.7%, while the second dominant genus *Pseudogracilibacillus* microorganisms in E2 and in OTC, increased by 66.0% and 40.7%, respectively, compared to those in the IUA control group (Figure 7G; Appendix A). Similar to the above phylum result, the E2 group showed the biggest difference from the IUA control group in species compositions, while the GXN group displayed little distinction from the IUA control group.

Further MetaStat analysis reveals that the proportions of *Ulvibacter* in the E2 group (2.04% in E2 vs. 0.55% in the IUA control, *p* = 0.03) and *Staphylococcus* in the OTC group (2.85% in OTC vs. 1.12% in the IUA control, *p* = 0.04) were significantly increased compared to those in the IUA control group (Figure 7H; Appendix A). The significant decrease in *Lactobacillus*, *Neisseria*, and *Prevotella* in OTC is consistent with a previous study in which OTC could inhibit 95% of *lactobacillus*, 80% of *Neisseria* and *Prevotella* [54,55].

A recent study reported that 97.1% of *Staphylococcus* spp. isolates from tonsils of slaughtered pigs displayed OTC resistance [56]. This was consistent with our results that OTC might not inhibit *Staphylococcus* microbiota in this study. However, up to now, there was no report about the functions of *Ulvibacter* in disease, or about the effect of OTC on the *Turicibacter* microorganism. The *Turicibacter* microorganism was only found in 12-week-old Tsumura, Suzuki, Obese, Diabetes mice that showed typical high-fat diet-independent type 2 diabetes symptoms, and was assumed to play an important role in the abnormal metabolism of type 2 diabetes [57].

Because the proportion of the *Ulvibacter* and IUA rate were both significantly increased in the E2 group, it seemed like there was a positive association between *Ulvibacter* and IUA, which was induced by the administration of E2. Similarly, there might also be a positive relationship between *Staphylococcus* and the IUA rate since the proportion of *Staphylococcus* and the IUA rate were both significantly increased in the OTC group. This result was consistent with the above result that the *Staphylococcus aureus* infection was a highly enriched KEGG pathway in IUA and a previous study which found that the *Staphylococcus* microorganism could acutely exacerbate pulmonary fibrosis [46].

Recent research on *Lactobacillus* has been contradictory. One suggested that administration of *Lactobacillus sakei* could aggravate bile duct ligation-induced liver inflammation and fibrosis in mice [58], while another indicated that *Lactobacillus rhamnosus* GG could reduce the hepatic fibrosis in a model of chronic liver disease in rats [59]. The *Prevotella* abundance was related to Th17 mediated mucosal inflammation, and *Prevotella* significantly drove Th17 immune response in vitro [60]. The maternal carriage of *Prevotella copri* during pregnancy strongly predicted the absence of food allergy in offspring [61].

### 3.9. Effects of E2, OTC and GXN on the Functions of Rat IUA Microbiota

A total of 6445 genes present in 19 samples were identified in four groups, E2 (n = 5), OTC (n = 6), GXN (n = 6) and IUA control (n = 6) groups (Figure 8A). Compared to the control group, 562, 725, and 93 genes significantly differed in abundances in the E2, OTC, and GXN groups, respectively (*p* < 0.05, Figure 8B,C). There were 43 up-regulated and 35 down-regulated microbial EDGs in E2, 12 up-regulated and 265 down-regulated microbial EDGs in the OTC group, and 2 up-regulated and 25 down-regulated microbial EDGs in GXN, respectively (*p* < 0.05, Fc > 2, Figure 8D,F; Appendix A). It seems that E2 displays the biggest effect, and GXN has the smallest effect, on the up-regulation of microbial EDGs, while OTC exhibits the strongest effect on the down-regulation of microbial EDGs.

Only two genes, K09935 assigned as a hypothetical protein and K03534 assigned as L-rhamnose mutarotase both involved in ko00001, were found to be significantly up-regulated in all the three groups (E2, OTC, and GXN) while 23 genes that were mainly involved in a two-component system, sulfur metabolism, metabolic pathways, microbial metabolism in diverse environments, and ko00001 pathways, were significantly down-regulated in all three groups (*p* < 0.05). Among all the significant differentially regulated genes, only one gene (assigned to ascorbate PTS system EIIC component, KO3475) that was found to be highly enriched in IUA compared to the non-IUA control before treatment, was significantly down-regulated in E2 compared to the IUA control after treatment.

The top three up-regulated pathways in E2 compared to the IUA control were tight junction, cell cycle, and HTLV-I infection but with less abundances than other significantly up-regulated pathways in E2, including p53 signaling pathways, protein digestion and absorption, apoptosis, renin angiotensin system, linoleic acid metabolism, and Chagas disease (American trypanosomiasis). The most down-regulated pathways in E2 were lipid metabolism, quorum sensing, protein kinases, transcription related proteins, epithelial cell signaling in *Helicobacter pylori* infection, mRNA surveillance pathway, sesquiterpenoid and triterpenoid biosynthesis, metabolism of xenobiotics by cytochrome P450, ABC transporters, and drug metabolism cytochrome P450 (Figure 8G). As expected, the two IUA highly enriched pathways before treatment, ABC transporters and quorum sensing that were both positively correlated with *M. pulmonis* (Figure 3C), were significantly down-regulated in the E2 group compared with IUA control after treatment (Figure 8G). It is interesting to note that one IUA highly enriched pathway before treatment, linoleic acid metabolism (Figure 4C,D), was further significantly up-regulated in the E2 group compared with the IUA control after treatment. Among all the significant differentially enriched pathways in OTC and GXN compared to the IUA control (Figure 8H,I; Appendix A), only arginine and proline metabolism, which was enriched in IUA compared to the non-IUA control with metabolome analysis (Figure 4C,D), was further up-regulated in the OTC group. There were no significant differentially enriched pathways in GXN (Figure 8H,I; Appendix A).

### 3.10. Effects of E2, OTC and GXN on the Rat IUA Endometrial Transcriptome

Endometrial transcriptome analysis displayed that there were 98 up-regulated and 81 down-regulated DEGs in the E2 group; 80 up-regulated and 69 down-regulated DEGs in the OTC group; and 314 up-regulated and 159 down-regulated genes in the GXN group (each group n = 4, *p* < 0.05, Fc > 2) (Figure 9A; Appendix A). Treatment with GXN yielded greater changes in gene transcription than treatment with E2 or OTC.

Among all the DEG in these three IUA endometrial groups after treatment, no DEGs were highly enriched in IUA compared to the non-IUA control before treatment while 10 DEGs were significantly up-regulated in all three treatment groups (E2, OTC, and GXN) (each group n = 4, *p* < 0.05, Fc > 2). Commonly up-regulated DEGs included *Ka11*, assigned to type I keratin involved in estrogen signaling pathway and *Staphylococcus aureus* infection; and *Caln1* assigned to calneuron 1 and 8 unknown genes (Figure 9B,D; Appendix A). Eleven significantly down-regulated DEGs were found in all three groups (*p* < 0.05, Fc > 2), including three unknown DEGs. The other eight DEGs included interleukin 6 (Il6); putative neuroendocrine convertase 1-like (AABR07024139.1); granzyme B (Gzmb); amine oxidase/copper containing 1 (Aoc1); tubulin folding cofactor B (Tbcb); putative snoRNA (AC114512.2); intercellular adhesion molecule 4/Landsteiner-Wiener blood group (Icam4); tissue factor pathway inhibitor 2 (Tfpi2); erythrocyte membrane protein band 4.1-like 1 (Epb41l1); and cysteinyl leukotriene receptor 2 (Cysltr2) involved in Calcium signaling pathways (Figure 9B,D; Appendix A).

Among these DEGs, Il-6 was dramatically down-regulated with decrease rates of 95.80% in E2, 85.62% in OTC, and 95.59% in GXN, respectively (Figure 9B,D; Appendix A). Il-6 is known to be involved in several pathways, including IL-17 signaling, TNF signaling, malaria, viral protein interaction with cytokine and cytokine receptor, NOD-like receptor signaling, African trypanosomiasis, Legionellosis, Cytokine–cytokine receptor interaction, Amoebiasis, Calcium signaling, and Rheumatoid arthritis [62]. A previous study indicated that IUA patients display significantly higher IL-6 level in serum than non-IUA patients [28]. However, in this study, the transcription levels of the IL-6 gene were decreased in the two of the IUA treatment groups (E2 and OTC), despite the fact that IUA in these two groups was significantly increased.

Compared to the IUA control the most up-regulated pathways in E2 endometria included the estrogen signaling pathway and taste transduction while the most down-regulated pathways included the IL-17 signaling pathway, TNF signaling pathway, malaria, NF-κB signaling pathway, viral protein interaction with cytokine and cytokine receptor, NOD-like receptor signaling pathway, Legionellosis, and cytokine–cytokine receptor interaction (Figure 9E,F, Appendix A). The inflammatory factor NF-κB expression that was significantly up-regulated in IUA endometria before treatment, was significantly down-regulated in E2 endometria. Surprisingly, the down-regulation of the inflammatory factor NF-κB signaling pathway, that was reportedly related to IUA formation [9], did not lead to a decrease of IUA in the E2 group.

Compared to the IUA control, the most up-regulated pathways in the OTC group were osteoclast differentiation and Herpes simplex virus 1 infection, while malaria, African trypanosomiasis, Renin-angiotensin system, TNF signaling pathway, and IL-17 signaling pathway, were the most down-regulated pathways in OTC endometria (Figure 9G,H; Appendix A). The increase in IUA after treatment with OTC might be ascribed to the up-regulation of some pathways, including B cell receptor signaling, Natural killer cell mediated cytotoxicity, NOD-like receptor signaling, C-type lectin receptor signaling, cytokine–cytokine receptor interaction, and Fc gamma R-mediated phagocytosis, all of which are involved in the immune system. Moreover, OTC, like E2, increased the high up-regulation of the estrogen signaling pathway in IUA endometria.

Compared to the IUA control, the most up-regulated pathway in the GXN group was the Herpes simplex virus 1 infection pathway, while IL-17 signaling pathway, malaria, African trypanosomiasis, TNF signaling pathway, viral protein interaction with cytokine and cytokine receptor, and cytokine–cytokine receptor interaction, were the most down-regulated in GXN endometria (Figure 9I,J; Appendix A).

The malaria pathway that had been significantly up-regulated in IUA compared to the non-IUA control before treatment, was significantly down-regulated in all three groups, E2, OTC and GXN. Two other pathways, cytokine–cytokine receptor interaction and viral protein interaction with cytokine and cytokine receptor, which had been significantly up-regulated in IUA compared to the non-IUA control before treatment, were significantly down-regulated in both the E2 and GXN groups. However, pathway B-Cell receptor signaling that had been significantly up-regulated in IUA compared to the non-IUA control before treatment, was further significantly up-regulated in both the E2 and OTC groups.

Previous research has suggested that *M. pulmonis* activates both rat B and T lymphocytes [63]. This mitogenic stimulation is more strongly effective upon B cells and less so on T cells. In our current study, despite the decrease in *M. pulmonis* levels in all the treatment groups, B-Cell receptor signaling was still up-regulated in the E2 and OTC groups.

## 4. Discussion

The association between multiple trauma and *Mycoplasmopsis* infection is well-known. It is assumed that the trauma might give rise to *Mycoplasmopsis* spp. Our results that *M. pulmonis* infection occurred in the traumatized endometrium is consistent with this specific association. The non-IUA horn shared the same dominant *M. pulmonis* with the IUA horn, most likely because the non-IUA horn was connected to the IUA horn, allowing *M. pulmonis* to spread and affect the environment, which could also explain why *M. pulmonis* was not detected in the intact and healthy uteri. However, the distinct growth rate and abundance of *M. pulmonis* between non-IUA horns and IUA horns caused by the trauma could allow us to distinguish *M. pulmonis* as the key microbial factor for IUA.

Though *M. pulmonis* predominated both the uterine horns in the non-IUA and IUA groups, the non-IUA horns without the trauma to the endometrium were still healthy without IUA while the horn with the trauma to the endometrium formed IUA, consistent with the previous study which found that the traumatic injury to the endometrium was an indispensable factor for IUA. The previous study indicated that the mechanical injury on the uteri without the damage to the endometrium, as a sham control, could not induce IUA and shared almost the same transcription of all the target genes with the other intact and healthy uteri controls. However, if the sham control can induce the *M. pulmonis* infection still remains unknown.

*M*. *pulmonis* is a mesophilic animal pathogen that is isolated from rat lung lesions. *M. pulmonis* infection has been shown to induce lung fibrosis in immunocompetent rat strains with no pre-existing lung disease as well as airway fibrosis in rat strains with differential susceptibility [64,65]. This was the first report of an association between *M. pulmonis* and rat IUA. It was found that the symptoms of lung diseases in mice caused by *M. pulmonis* were quite similar to those in human-by-human lung pathogen *M. pneumonia*, which induced chronic respiratory infection, airway hyperreactivity, pulmonary inflammation, and lung fibrosis [64].

A previous study reported that *M. genitalium* is an important cause of pelvic inflammatory disease and acute endometritis [22,23]. We thus compared the metabolic profiles of uterine flushing fluids from IUA patients and endometritis patients and found out the possible role of *Mycoplasmopsis* involvement in human IUA based on the similar IUA-associated metabolites between humans and rats. Moreover, the distinct metabolic profiles of two uterine flushing fluids between human IUA and endometritis indicates that the endometritis-associated *M. genitalium* should not be the microbe for IUA formation in humans. Most likely, the human pathogen *M**. pneumonia*, which causes lung disease and fibrosis in humans, might be involved in IUA formation. This might explain why pelvic inflammatory disease and endometritis could enhance the occurrence and recurrence of IUAs likely ascribed to some similar characteristics between *M. pneumonia* and *M. genitalium* [35,64,65].

Transcriptional analysis of endometria revealed that 60% of the top 20 IUA-enriched DEGs were involved in immune responses and lymphocyte activation, consistent with previous reports that *Mycoplasmopsis* interacted with the host immune system and that their presence was mitogenic for lymphocytes [48,66]. Tenascin N, which was involved in allowing cell rounding and promoting condensation [37,38,39] and might play a key role in IUA formation, was highly induced in IUA endometria. This was the first time that Tenascin N was related to fibrosis. Moreover, the genes involved in the production of leydig and trophectoderm cells, and in the immortalization of activated T cells and monocytes, respectively, were all significantly down-regulated in IUA endometria. In summary, multiple predisposing and causative factors associated with IUA formation were strongly related to *M. pulmonis*.

Surprisingly, treatment with E2 and OTC did not reduce the IUA nor increase endometrial glands as expected. Instead, in both the E2 and OTC groups, IUA became worse. Meanwhile, the level of TGF-β1, which was involved in IUA formation, was much higher than that in the control group. The lack of reduction in fibrosis in endometria with E2 treatment was inconsistent with previous studies that have shown a reduction in fibrosis in different organs, including cardiac, liver, lung, and endometria, with E2 treatment [15,16,17,18]. However, our result was consistent with another study in which estrogen increased the high expression of Tenascin associated with fibrosis formation in human subjects [67]. The low number of rat IUA models in the medicine treatment could lead to biased results. Taking more samples at several time points during IUA formation and the following medicine treatment will help better analysis.

The possible association between an IUA increase and treatments with E2 and OTC, add another dimension to our understanding of IUA reoccurrence. It seems that *Ulvibacter* and *Staphylococcus* could also enhance IUA, because the proportion of *Ulvibacter* with IUA rate and of *Staphylococcus* with IUA rate were both significantly increased in E2 and OTC groups, respectively, partly consistent with a previous study in which *Staphylococcus* microorganism induced acute exacerbation of pulmonary fibrosis [46]. Further evidence comes from the fact that *Staphylococcus aureus* infection was a highly IUA-enriched KEGG pathway in endometria in our study. Moreover, two up-regulated pathways in E2 and OTC treatments were the IUA-enriched B cell receptor signaling [68], and the estrogen signaling pathway, which warrants further investigation for their involvement in IUA.

In conclusion, our findings extend insights into the relationship of IUA formation with intrauterine microbiota and endometrial factors and provide a basis for the development of new strategies and therapy for IUA treatment.

## Figures and Tables

**Figure 1 pathogens-11-00784-f001:**
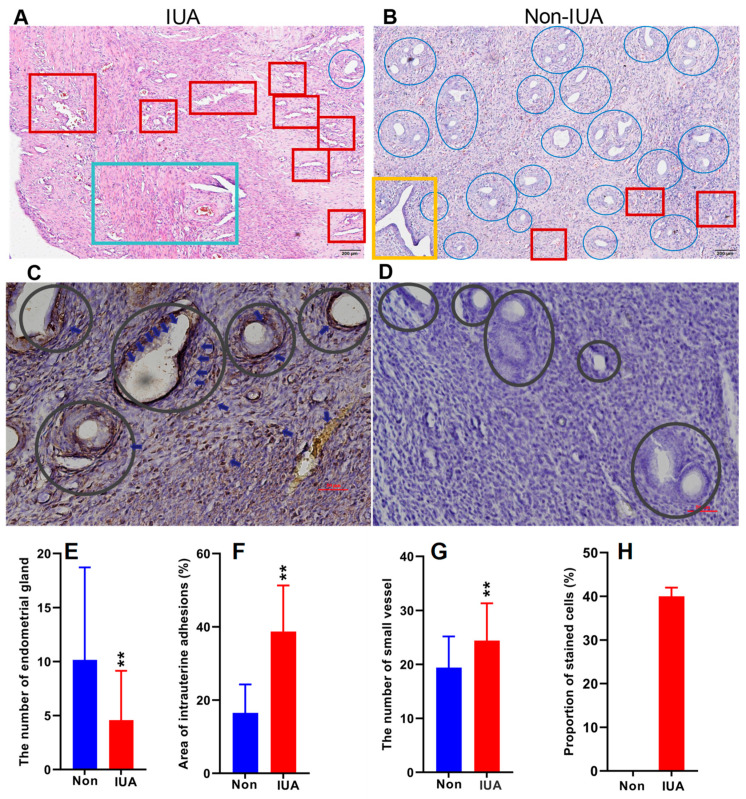
Comparison of the hematoxylin and eosin (H&E) stained endometria (Bar = 200 μm) (**A**,**B**) and immunohistochemical reactivity (Bar = 50 μm) (**C**,**D**) for TGF-β1 in IUA and non-IUA control endometria. Endometrial glands in blue circle and expanded endometrial glands at the secretion stage in yellow rectangle; endometrial fibrosis in green rectangle; small vessel infiltration in red rectangle; (**E**) Comparison of the diagnostic indices between IUA and non-IUA control groups, including endometrial glands (**E**); micro-vessels (**F**); average fibrotic area (**G**); and stained cells (**H**); (Student’s *t*-test. ** *p* < 0.01).

**Figure 2 pathogens-11-00784-f002:**
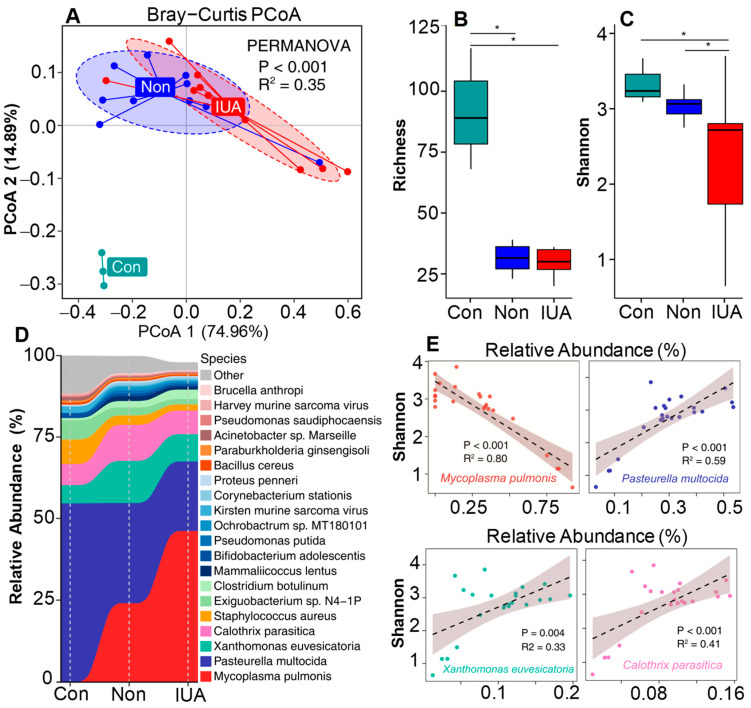
The effect of *Mycoplasm**opsis pulmonis* infection in IUA rats on inter-uterine dysbacteriosis. (**A**) Comparison of microbial community composition differences among control (Con) rats, Non-IUA (Non) horns and IUA horns based on the relative abundance at species level by using Principal coordinate analysis (PCoA) based on Bray Curtis dissimilarity distance. Significant differences test between groups were computed using PERMANOVA. (**B**,**C**), Comparison of the richness and Shannon’s diversity of Con, Non, and IUA groups. BH-corrected *p* value computed from Wilcoxon test (* *p* < 0.05). (**D**) Relative abundance of the top 25 abundant bacterial species in microbial communities of Con, Non, and IUA groups. (**E**) A strong negative association of *M**. pulmonis* with microbial diversity by comparison of the Pearson correlation between Shannon diversity indices and the relative abundance of top four dominant bacterial species.

**Figure 3 pathogens-11-00784-f003:**
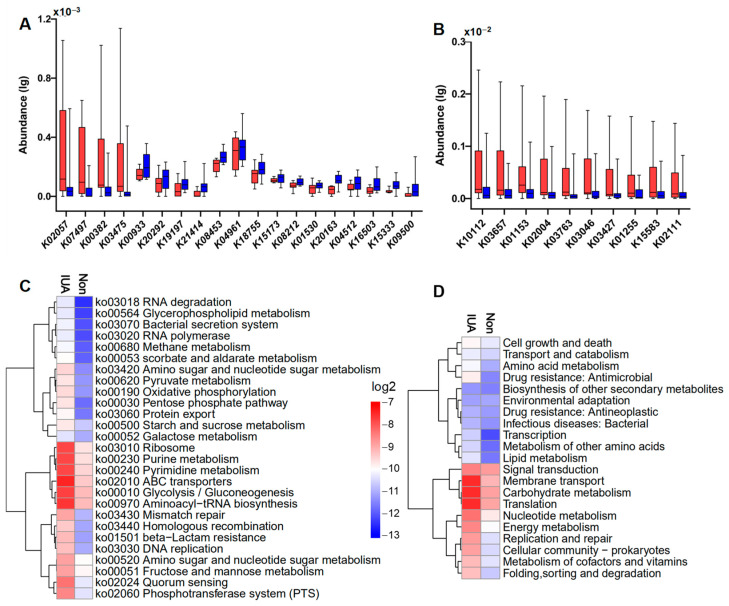
(**A**) Significantly differential regulations of indicated microbial KOs in IUA (red) vs. non-IUA control (blue) (control, n = 46; IUA, n = 46; Metastats analysis, *p* < 0.05). Up-regulated Kos in the IUA group were simple sugar transport system permease protein K02057; putative transposase K07497; dihydrolipoamide dehydrogenase K00382; ascorbate PTS system EIIC component K03475. Down-regulated Kos in IUA were ryanodine receptor 1 K04961; adhesion G-protein coupled receptor F1 K08453; importin-8 K18755; creatine kinase K00933; transcription termination factor 2 K15173; conserved oligomeric Golgi complex subunit 5 K20292; OCT family organic cation transporter K08212; phospholipid-translocating ATPase K01530; domain-containing protein K20163; inhibitor of growth protein 1 K19197; disheveled associated activator of morphogenesis K04512; cadherin-related family member 3 K16503; tRNA guanosine-2′-O-methyltransferase K15333; ankyrin repeat and SAM domain-containing protein 4B K21414; T-complex protein 1 subunit theta K09500. (**B**) Significant differential enrichment of the top ten abundant indicated microbial Kos in IUAs (red) compared to those in controls (red) (IUA, n = 46; control, n = 46; *t*-test, *p* < 0.01). (**C**) Hierarchical clustering and heat map showing relative abundance (log2 scale) of differential pathways (level 3) in IUA and non-control. The top ten abundant microbial pathways in the IUA group were: ribosome, purine metabolism, pyrimidine metabolism, ABC transporters, glycolysis/gluconeogenesis, aminoacyl−tRNA biosynthesis, amino sugar and nucleotide sugar metabolism, fructose and mannose metabolism, quorum sensing, and phosphotransferase system (PTS), whose enrichments were significantly higher than those in the non-IUA group. (Control, n = 46; IUA, n = 46; *t*-test, *p* < 0.01). (**D**) Hierarchical clustering and heat map showing relative abundance (log2 scale) of the differentially regulated microbial pathways (level 2) in IUA vs. non-IUA. The top 10 abundant microbial pathways in the IUA group were signal transduction, membrane transport, carbohydrate metabolism, translation, nucleotide metabolism, energy metabolism, replication and repair, cellular community-prokaryotes, metabolism of cofactors and vitamins, whose enrichments were significantly higher than those in the non-IUA group. (Control, n = 46; IUA, n = 46; *t*-test, *p* < 0.01).

**Figure 4 pathogens-11-00784-f004:**
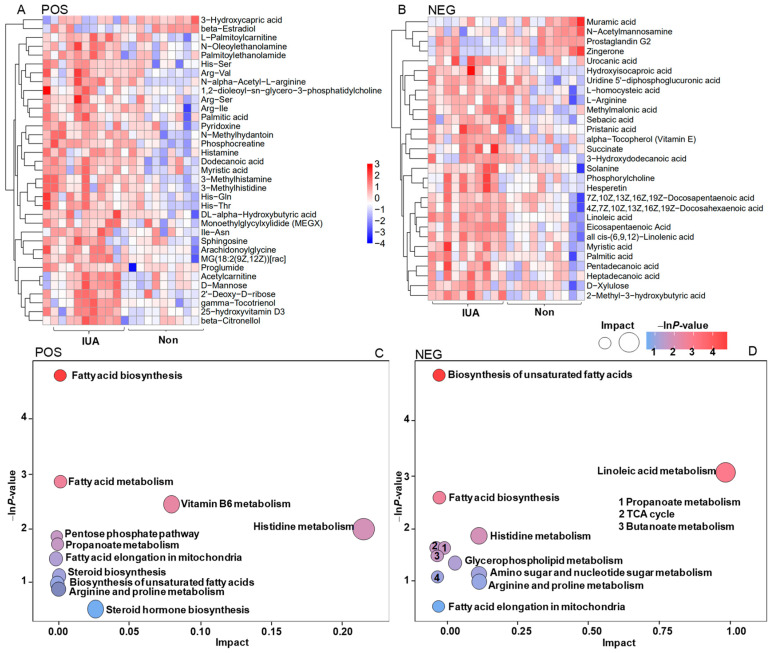
Heatmap with Hierarchical clustering for significantly differential enrichment of uterine metabolites (**A**,**B**) and KEGGs (**C**,**D**) in controls and IUAs and associations of IUA microbial species with circulating metabolites (**E**,**F**). (**A**,**B**) Heatmap with Hierarchical clustering for differentially regulated metabolites in IUA vs. non-control uteri identified with MS in positive ion mode MS (POS, (**A**)) and in negative ion mode (NEG, (**B**)). (**C**,**D**) The differentially regulated KEGG pathways in IUA and non-control uteri based on that these differentially regulated uterine metabolites in IUA vs. non-control uteri identified with POS I and NEG (**D**). (**E**,**F**) Spearman’s correlation between the strains including *Mycolasma pulmonis*, *Oceanospirillum multiglobuliferum*, *Chlamydia abortus*, *Cyberlindnera jadinii*, *Piromyces finnis*, *Mircobactium profundi* and metabolites that differed in abundance between IUA and non-IUA uteri (POS: (**E**), NEG: (**F**), *: *p* < 0.05).

**Figure 5 pathogens-11-00784-f005:**
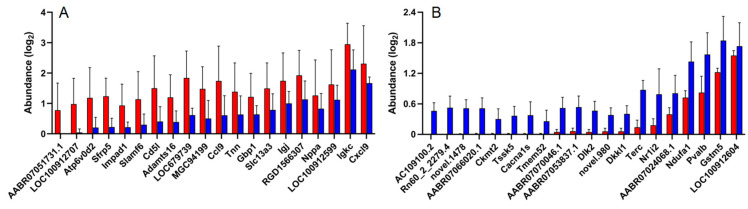
(**A**) The top 20 up-regulated genes in IUA endometria vs. non-IUA control. (IUA, n = 6; control, n = 6; Fold change > 3, *t*-test *p* < 0.01). (**B**) The top 20 down-regulated genes in IUA endometria vs. non-IUA control. (IUA, n = 6; control, n = 6; Fold change < −3 and *t*-test *p* < 0.01). (**C**,**D**) The up-regulated (**C**) and down-regulated (**D**) KEGG pathways in IUA endometria vs. non-IUA control. Only the top 20 annotations are shown. The most up-regulated genes in IUA endometria compared to the non-IUA control are involved in the Cytokine–cytokine receptor interaction, Chemokine signaling pathway, *Staphylococcus aureus* infection, Hematopoietic cell lineage, Natural killer cell mediated cytotoxicity, Complement and coagulation cascades, B cell receptor signaling pathway, Th17 cell differentiation, and T cell receptor signaling pathway, while the most down-regulated genes are involved in the Ribosome.

**Figure 6 pathogens-11-00784-f006:**
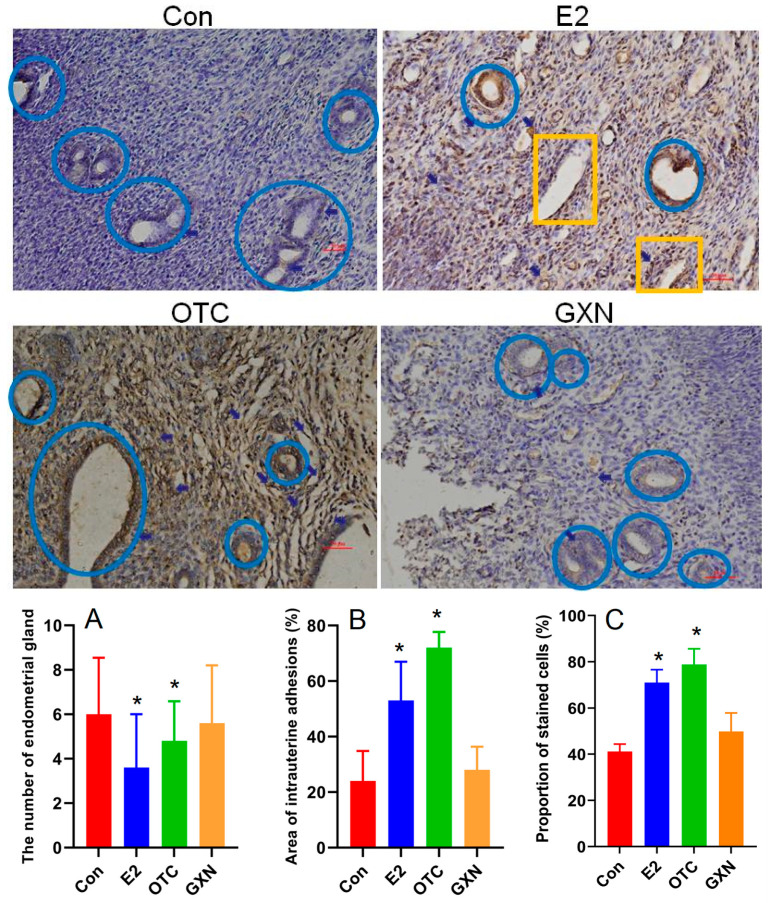
Comparison of the hematoxylin and eosin (H&E) stained endometria (Bar = 50 μm) and immunohistochemical reactivities (Bar = 50 μm) for TGF-β1 in E2, OTC, GXN and IUA control endometria (E2, n = 5; OTC, n = 6; GXN, n = 6; control n = 6). Endometrial glands in blue circle; small vessels in yellow rectangle. (**A**–**C**) Comparison of the diagnostic indices between the four groups, (**A**) endometrial glands; (**B**) IUA area; (**C**) TGF-β1 expression level (Student’s *t*-test. * *p* < 0.05).

**Figure 7 pathogens-11-00784-f007:**
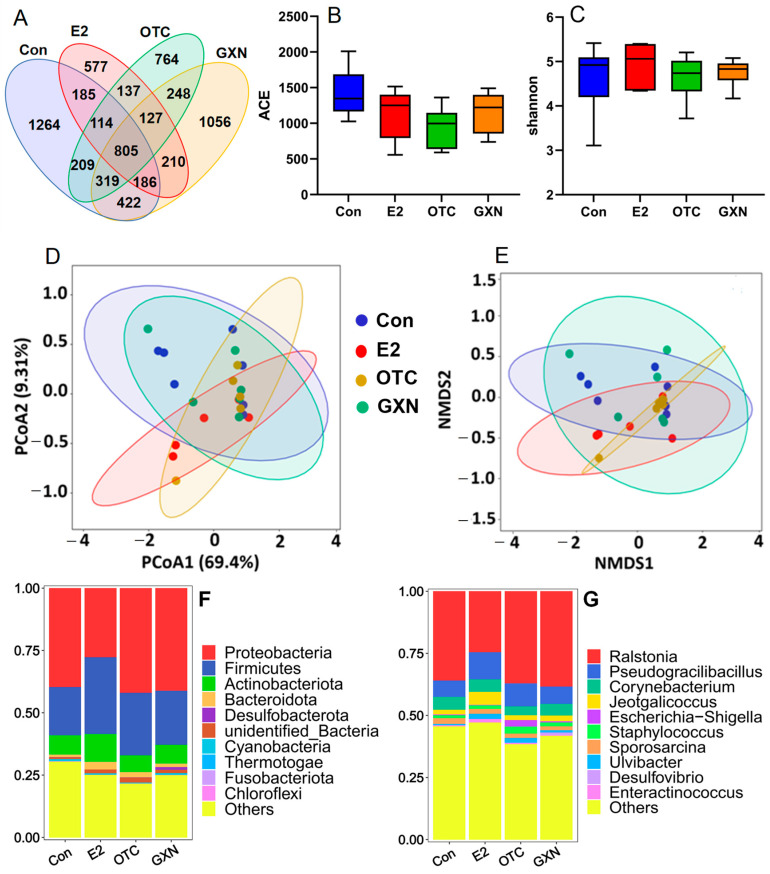
(**A**) Venn analysis of microbiota in four groups, E2, OTC, GXN and IUA control with 16s rRNA sequencing (E2, n = 5; OTC, n = 6; GXN, n = 6; IUA control n = 6). (**B**,**C**) Alpha-diversity analysis, including ACE (**B**) and Beta-diversity analysis, including Shannon index (**C**), of microbiota in four groups, E2, OTC, GXN and IUA control. (**D**,**E**) PCoA and NMDS analysis for the distribution and composition of species in the four groups, E2, OTC, GXN and IUA control. (**F**,**G**): Comparison of microbial compositions of the top 10 enriched microorganisms among the four groups at the phylum (**F**) and genus (**G**) levels. (**H**) Significantly differential enrichment of species in the E2, OTC, and GXN groups compared to the IUA control group, respectively.

**Figure 8 pathogens-11-00784-f008:**
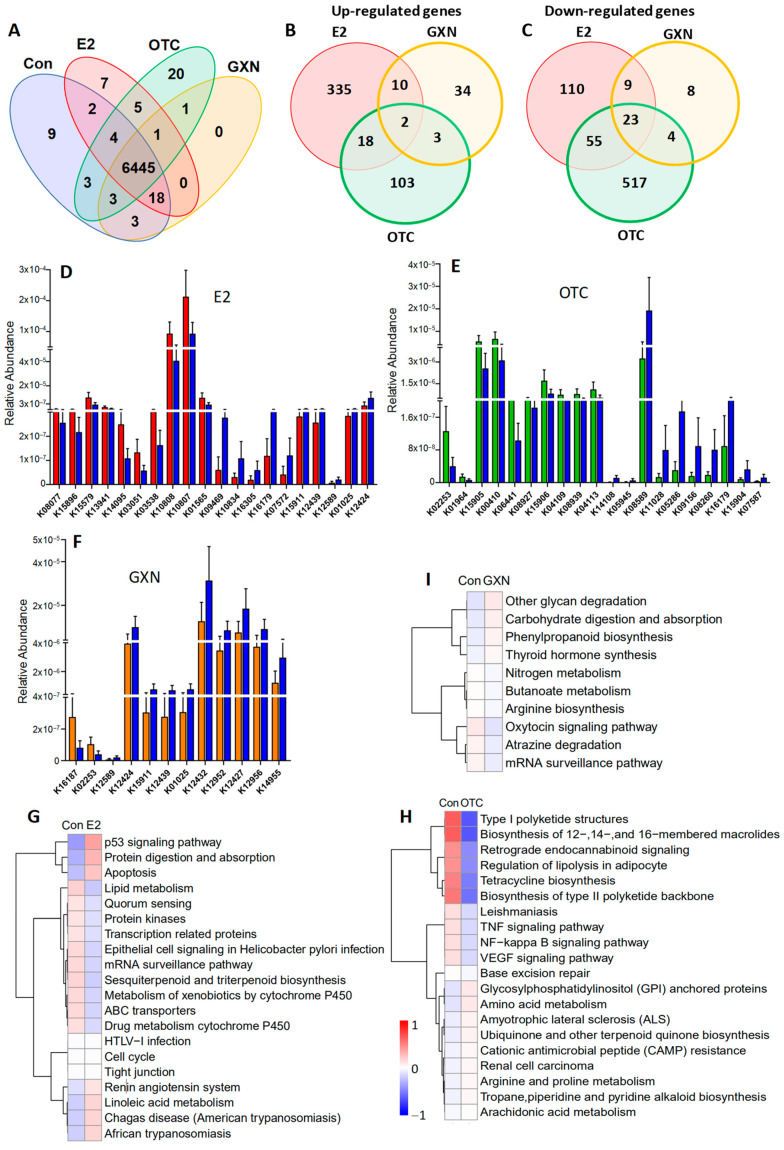
(**A**) Venn analysis of the indicated microbial KO numbers among the four different groups (E2, n = 5; OTC, n = 6; GXN, n = 6; IUA-Con, n = 6). (**B**,**C**) Venn analysis for the numbers of significantly up-regulated (**B**) and down-regulated KOs (**C**) in E2, OTC and GXN groups, compared to the IUA control group (E2, n = 5; OTC, n = 6; GXN, n = 6; IUA control, n = 6). (**D**) The top 10 up-regulated and 10 down-regulated KOs (red) in E2 compared to those in IUA controls (blue) (E2, n = 5; control n = 6, <0.05 and fold change > 2). (**E**) The top 10 up-regulated and 10 down-regulated KOs (red) in OTC compared to those in IUA controls (blue) (OTC, n = 6; control n = 6, *p* < 0.05 and fold change > 2). (**F**) The top 10 up-regulated and 10 down-regulated KOs (red) in GXN compared to those in controls (blue) (GXN, n = 6; control n = 6, *p* < 0.05). (**G**) KEGG analysis of categorizing all the significantly up-regulated and down-regulated genes in the E2 group compared to IUA control (relative abundance, log2 scale, *p* < 0.05). Only the top 20 changed pathways are shown. (**H**) KEGG analysis of categorizing all the significantly up-regulated and down-regulated genes in the OTC group compared to control (relative abundance, log2 scale, *p* < 0.05). Only the top changed 20 annotations are shown. The most up-regulated pathways in OTC compared to IUA control were glycosylphosphatidylinositol (GPI) anchored proteins, amino acid metabolism, amyotrophic lateral sclerosis (ALS), ubiquinone and other terpenoid quinone biosynthesis, cationic antimicrobial peptide (CAMP) resistance, renal cell carcinoma, arginine and proline metabolism, Tropane/piperidine and pyridine alkaloid biosynthesis, arachidonic acid metabolism. The most down-regulated pathways in OTC were base excision repair, NF-kappa B signaling pathway, VEGF signaling pathway, TNF signaling pathway, leishmaniasis, tetracycline biosynthesis, retrograde endocannabinoid signaling, regulation of lipolysis in adipocyte, Type I polyketide structures, biosynthesis of 12-, 14-, and 16-membered macrolides, Biosynthesis of type II polyketide backbone. (**I**) KEGG analysis of categorizing all the significantly up-regulated and down-regulated genes in GXN compared to IUA control. The most up-regulated pathways in the GXN group compared to the IUA control were other glycan degradation, carbohydrate digestion and absorption, phenylpropanoid biosynthesis, thyroid hormone synthesis, while the most down-regulated pathways were nitrogen metabolism, butanoate metabolism, arginine biosynthesis, oxytocin signaling pathway, atrazine degradation, and mRNA surveillance pathway.

**Figure 9 pathogens-11-00784-f009:**
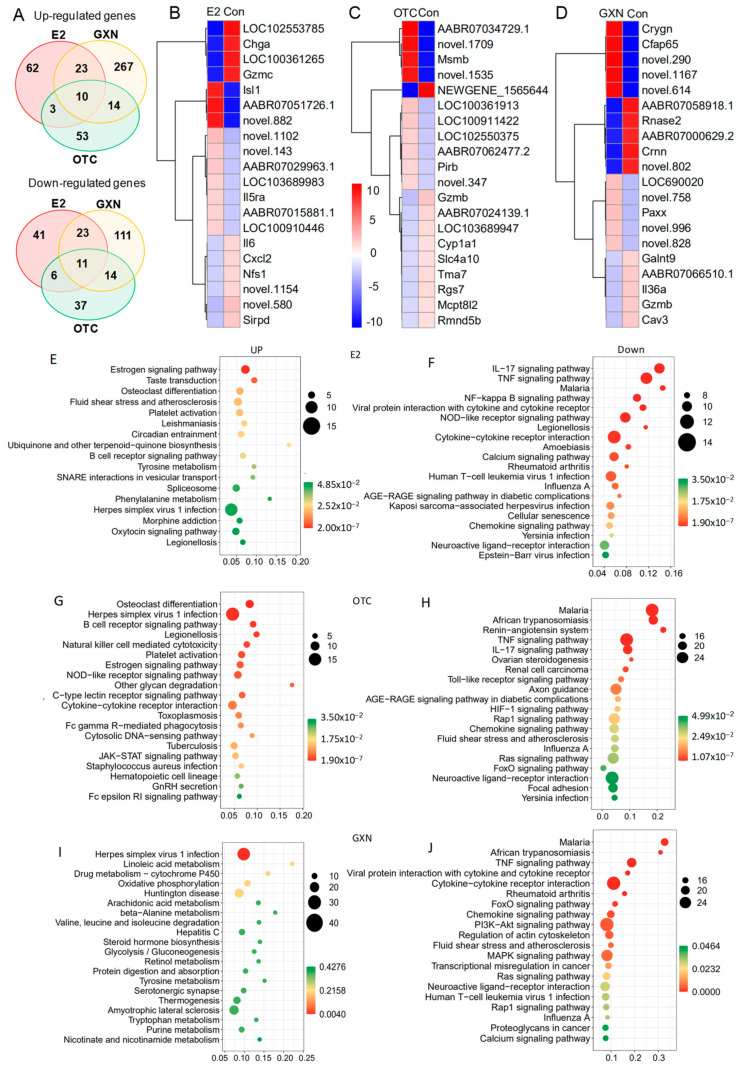
(**A**) Venn analysis for the numbers of significantly up-regulated and down-regulated genes in E2, OTC and GXN endometria, compared to the IUA control group (E2, n = 4; OTC, n = 4; GXN, n = 4; IUA control, n = 4). (**B**) The top 10 up-regulated and 10 down-regulated genes in E2 compared to those in IUA controls (E2, n = 4; control n = 4, *p* < 0.05 and Fc > 2). (**C**) The top 10 up-regulated and 10 down-regulated genes in OTC compared to those in IUA controls (OTC, n = 4; IUA control n = 4, *p* < 0.05 and Fc > 2). (**D**) The top 10 up-regulated and 10 down-regulated genes in GXN compared to those in IUA controls (GXN, n = 4; control n = 4, *p* < 0.05 and Fc > 2). (**E**,**F**) KEGG analysis of categorizing all the significantly up-regulated and down-regulated genes in the E2 group compared to the IUA control (relative abundance, log2 scale, *p* < 0.05). Only the top 20 changed pathways are shown. (**G**,**H**) KEGG analysis of categorizing all the significantly up-regulated and down-regulated genes in the OTC group compared to the IUA control (relative abundance, log2 scale, *p* < 0.05). Only the top changed 20 annotations are shown. (**I**,**J**) KEGG analysis of categorizing all the significantly up-regulated and down-regulated genes in GXN compared to IUA control.

## Data Availability

All the data and methods necessary to reproduce this study are included in the manuscript and Appendix A. 16S rRNA amplicon dataset is available in the NCBI SRA repository (PRJNA781379, https://www.ncbi.nlm.nih.gov/bioproject/PRJNA781379; accessed on 4 March 2022); Metagenome dataset is available in the NCBI SRA repository (PRJNA781408, https://www.ncbi.nlm.nih.gov/bioproject/PRJNA781408; accessed on 4 March 2022); Transcriptome datasets collected in this study are available in the NCBI SRA repository (PRJNA783470, https://www.ncbi.nlm.nih.gov/bioproject/PRJNA783470; accessed on 4 March 2022); Metabolomics dataset is available in MetaboLights repository (MTBLS3867, https://www.ebi.ac.uk/metabolights/editor/guide/assays/MTBLS3867; accessed on 4 March 2022).

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
