# Peer review of "Association of Intrauterine Microbes with Endometrial Factors in Intrauterine Adhesion Formation and after Medicine Treatment"

_pathogens, 2022, doi:10.3390/pathogens11070784_

Round 1

Reviewer 1 Report

1. The study presents extensive evaluation of microbiome taxonomy, genomic, transcriptome and metabolic characteristics in the rat IUA model. However, there is an issue affecting the significance of the study that could be resolved by adding control groups to the experiments. If this is not feasible, the limitations should be stated explicitly.

Namely, the uterine horns from rats without surgery (“pre-control”, Con, lines 378–379) were not used as control group in most experiments. They only compared IUA and non-IUA horns from rats that underwent surgery. Yet non-IUA horn is connected to the IUA-horn, allowing the bacteria to spread and to affect their environment. The study thus demonstrates that some bacterial species prefer the scarred compartments in the same system. This sort of comparison doesn’t involve intact and healthy uterus. The particular findings exemplify the problem.

 Mycoplasmopsis pulmonis was abundant in IUA rat model (less so in the non-IUA uterus horn) yet absent from pre-control rat uteri, according to shotgun sequencing data of the microbiomes (Fig. 2, Table S5). Microbiomes of pre-control group significantly differed from both IUA and non-IUA horns (Fig. 2A). This find confirms that IUA and non-IUA comparison leaves out the ‘healthy’ condition.

Also, the study discovered that after three weeks of treatment with three medicines or distilled water, M. pulmonis was no longer dominant (lines 660–664). An experiment demonstrating that M. pulmonis is not introduced because of any peculiarity of the protocol would be very informative. The study that is referenced in the current article [1] used sham operation without destruction of the endometrium. Careful insertion of the needle without scraping could mimic the intervention very closely without causing IUA. This sort of control group could allow to judge whether the changes in microbiota were not caused by intervention or healing conditions independently from developing IUA.

 2. Introduction would benefit from editing for clarity. Also, the lines 60–62 ought to be rewritten; line 70: immune system, not ‘systems’; line 93: CE wasn’t introduced as an abbreviation before.

 3. The caption of Fig. 1 (lines 344, 345) should say H&E stained endometria is in A&C, while immunohistochemical staining is in B&D. The caption of Fig. 4 in line 502 should say ‘IUA vs. non-IUA’.

4. M. pulmonis is called Mycoplasmopsis or Mycoplasma (lines 28, 396, 496) interchangeably in this study. Please use the latest form of the nomenclature consistently to avoid the reader’s cofusion.

 5. The chapter 3.5 (lines 509–537) examines the metabolites washed out from human uteri in the presence of disease, including IUA. The experimental setup allows for comparison among different conditions. However, it is not stated that any microbiome investigation was carried out in these patients. There is only literature reference (line 512) about the role of Mycoplasma genitalium in human endometritis, yet no information if any particular species was present in the genesis of the diseases in this study. Thus the conclusions about the bacterial species are not supported by data. There only may be a speculation.

 6. Discussion, lines 867–881: the role of Mycoplasma pneumonia in human disease is proposed, yet the Results have no data on its presence in human samples. The similarity of lung-associated symptoms between M. pulmonis and M. pneumonia is not enough to support the proposal. The hypothetical quality of this suggestion should be made more explicit.

 7. The study has very extensive analysis of the effects of three medications in rat IUA model, including changes in histology, microbiota, its metabolic pathways, and endometrial transcriptome. However, only 23 rats were used in this comparison (E2, n = 5; OTC, n = 6; GXN, n = 6; control n = 6; line 636). This detail should be mentioned in the Discussion together with other limitations of the current study.

Reference: 1. Guo LP, Chen LM, Chen F, Jiang NH, Sui L. Smad signaling coincides with epithelial-mesenchymal transition in a rat model of intrauterine adhesion. Am J Transl Res. 2019 Aug 15;11(8):4726-4737. PMID: 31497194; PMCID: PMC6731410.

Author Response

Dear Reviewer,

Thank you very much for taking time out of your busy schedule to read this article. Thank for providing many constructive modification comments and suggestions, which have greatly helped us to further improve this manuscript. We carefully thought and analyzed your valuable opinions. On this basis, we have made the appropriate revisions and responses to your comments and suggestions, as noted below.

  1. The study presents extensive evaluation of microbiome taxonomy, genomic, transcriptome and metabolic characteristics in the rat IUA model. However, there is an issue affecting the significance of the study that could be resolved by adding control groups to the experiments. If this is not feasible, the limitations should be stated explicitly. Namely, the uterine horns from rats without surgery (“pre-control”, Con, lines 378–379) were not used as control group in most experiments. They only compared IUA and non-IUA horns from rats that underwent surgery. Yet non-IUA horn is connected to the IUA-horn, allowing the bacteria to spread and to affect their environment. The study thus demonstrates that some bacterial species prefer the scarred compartments in the same system. This sort of comparison doesn’t involve intact and healthy uterus. The particular findings exemplify the problem. Mycoplasmopsis pulmoniswas abundant in IUA rat model (less so in the non-IUA uterus horn) yet absent from pre-control rat uteri, according to shotgun sequencing data of the microbiomes (Fig. 2, Table S5). Microbiomes of pre-control group significantly differed from both IUA and non-IUA horns (Fig. 2A). This find confirms that IUA and non-IUA comparison leaves out the ‘healthy’ condition.

Response: We entirely agree with the reviewer’s comment that an intact and healthy control in some experiments might help contribute to a comprehensive analysis of the microbial factor associated with IUA. Because the non-IUA horn is connected to the IUA-horn, which does allow the IUA associated microbes to spread and to affect their environment, and some microbial species that prefer the scarred compartments in the same system could be observed in our study. Thus, the non-IUA horn shared the same microbe with IUA-horn due to the spread of the IUA associated microbes and their affection. To be honest, we couldn’t do the perfect research for one time. In this study, what we could do was to find out a key microbial factor for IUA. Though the non-IUA horn shared the same microbial species with IUA, the distinct surgical treatments between two horns resulted in distinct growth rates and abundances of the microbes in these two different horns, which allowed us to distinguish the microbial factor. In order to avoid the possible false results, the association of the microbial factor with IUA was further elucidated by comprehensive analysis of the diagnostic characteristics, transcriptome and metabolic characteristics in the rat IUA and non-IUA models, and detailed comparison of the metabolic characteristics in human IUA and rat IUA. Indeed, the intact and healthy control was very important because it might provide some microbial species which might help maintain endometrial health and inhibit IUA. Thus, our next investigation will plan to analyze the beneficial microbial factors in uterine.

Also, the study discovered that after three weeks of treatment with three medicines or distilled water, M. pulmonis was no longer dominant (lines 660–664). An experiment demonstrating that M. pulmonis is not introduced because of any peculiarity of the protocol would be very informative. The study that is referenced in the current article [1] used sham operation without destruction of the endometrium. Careful insertion of the needle without scraping could mimic the intervention very closely without causing IUA. This sort of control group could allow to judge whether the changes in microbiota were not caused by intervention or healing conditions independently from developing IUA.

Response: Our result that after three weeks of treatment with three medicines or distilled water, M. pulmonis was no longer dominant, was consistent with previous studies that Mycoplasmopsis infection usually resolved spontaneously due to its self-resolving nature because Mycoplasmopsis would die on their own normally within 3 weeks [52,53]. Our experiment for the construction of IUA in rat model was totally carried out according to the commonly used protocol as described in a previous study [29], which involved protection procedures against microbial infection during the surgical performance on uterine. Moreover, 46 rats that were divided into 3 batches were used for the IUA models in our study. We believe that our finding that M. pulmonis is a key microbial factor for IUA, would help improve protection procedures against Mycoplasmopsis infection during the surgical performance on uterine when our study will be published. Our next investigation on IUA will focus on a specific protocol for inhibiting Mycoplasmopsis infection.

Before this study, we had carefully read the reference 1 (as ref 29 in our manuscript, Guo LP, Chen LM, Chen F, Jiang NH, Sui L. Smad signaling coincides with epithelial-mesenchymal transition in a rat model of intrauterine adhesion. Am J Transl Res. 2019 Aug 15;11(8):4726-4737. PMID: 31497194; PMCID: PMC6731410.), in which, a mechanical injury or sham operation was performed on the left uterus (IUA-L, Sham-L), and the right uterus (IUA-R, Sham-R) was used as the control. We had found out that the sham control shared almost the same with the other control. As it said in the abstract of ref 1: “In the IUA-L group, TGF-1 and pSmad3 levels were consistently high, and levels of BMP7, pSmad1/5 and ER were low. EMT markers E-cadherin was decreased, while N-cadherin was increased, but the sham operation group and control groups showed no significant difference in these markers.” Based on these results from ref 1, we assumed that the sham operation group should be quite similar to those of the other two control groups. Combined with the consideration in response to the first question, the sham operation group was not used in our study, also in order to reduce the number of surgical rats. We wish that the reviewer could understand our limitation. Our next investigation on IUA will plan to set up a sham control to achieve a more comprehensive analysis.

  1. Introduction would benefit from editing for clarity. Also, the lines 60–62 ought to be rewritten; line 70: immune system, not ‘systems’; line 93: CE wasn’t introduced as an abbreviation before.

Response: Introduction has been edited for clarity. The lines 60–62 have been rewritten. “systems” has been changed to “system”. “CE” has been added after “chronic endometritis” in the line before the abbreviation CE.

  1. The caption of Fig. 1 (lines 344, 345) should say H&E stained endometria is in A&C, while immunohistochemical staining is in B&D. The caption of Fig. 4 in line 502 should say ‘IUA vs. non-IUA’.

Response: Thanks. Here were our mistakes. Corrections in Figures 1 and 4 have been done.

  1. M. pulmonisis called Mycoplasmopsis or Mycoplasma (lines 28, 396, 496) interchangeably in this study. Please use the latest form of the nomenclature consistently to avoid the reader’s cofusion.

Response: The nomenclature has been unified in this manuscript.

  1. The chapter 3.5 (lines 509–537) examines the metabolites washed out from human uteri in the presence of disease, including IUA. The experimental setup allows for comparison among different conditions. However, it is not stated that any microbiome investigation was carried out in these patients. There is only literature reference (line 512) about the role of Mycoplasma genitaliumin human endometritis, yet no information if any particular species was present in the genesis of the diseases in this study. Thus the conclusions about the bacterial species are not supported by data. There only may be a speculation.

Response: We agree with the reviewer’s comments that the microbial investigation in these patients should be carried out. Because the individual difference of these patients could lead to distinct microbial data, a huge amout of samples should have been required for a better analysis. Furthermore, the sample quantity was insufficient for both analysis, so we chosed the metabolic profiles for the analysis of the difference between IUA and endometritis, because the individual difference of thepatients made less effect on the metabolic profile than on the microbial data (Nature https://doi.org/10.1038/s41586-022-04828-5). In order to better clarify the chapter 3.5, we changed the subtitle to “Similar metabolites between human IUA and rat IUA while distinct metabolic profiles between human IUA and human endometritis”. We highlighted that the IUA associated Mycoplasmopsis species characterized with the rat model was convincing.

  1. Discussion, lines 867–881: the role of Mycoplasma pneumoniain human disease is proposed, yet the Results have no data on its presence in human samples. The similarity of lung-associated symptoms between M. pulmonisand M. pneumonia is not enough to support the proposal. The hypothetical quality of this suggestion should be made more explicit.

Response: Our results from IUA rat model in this study revealed that M. pulmonis, a key microbial factor for murine lung fibrosis, was also involved in rat IUA. Combined with previous studies that M. pneumonia was a key causal microbial factor in human lung fibrosis, we ventured to deduce that human lung pathogen M. pneumonia might also be involved in human IUA. Indeed, this needs further investigation. We hope that our findings in rat model could be instructive and meaningful to human related disease. We sincerely wish that the reviewer could understand the limitation of our study.

  1. The study has very extensive analysis of the effects of three medications in rat IUA model, including changes in histology, microbiota, its metabolic pathways, and endometrial transcriptome. However, only 23 rats were used in this comparison (E2, n = 5; OTC, n = 6; GXN, n = 6; control n = 6; line 636). This detail should be mentioned in the Discussion together with other limitations of the current study.

Response: This limitations of the current study have been mentioned in the Discussion as following: The low number of rat IUA models in the medicine treatment could lead to biased results. Taking more samples at several time points during IUA formation and the following medicine treatment will help better analysis.

Reviewer 2 Report

There are many grammatical / translational errors that need to be addressed:

Line 42: change 'unveiled' to 'found'

Line 45: the cause factor should be causal factors, or contributing factors.

Different fonts sizes, such as in lines 56-58 need to be corrected throughout the manuscript

Lines 86-87: I think you need to put in (CE) following chronic endometritis

Lines 93-94: ‘have’ not ‘has’… and ‘of’ not ‘in’…  look for many similar errors

Line 102: change pouch of Douglas to recto-genital pouch

Lines 99-108: I suggest changing ‘canal’ to ‘lumen’

Lines 109-110: However, they do not have a duplex cervix. What are the risks of cross migration / contamination?

Line 119: OTC has known tissue toxicity, this is a harsh product to use.

Line 145: another common error that needs to be corrected throughout – ‘the’ needs to be inserted between operation on ‘the’ uteruses… actually uteri is the proper term. There are many such instances throughout the manuscript, I use this as one example.

Line 146: scrapped down is confusing. A more complete description of how the endometrium was obtained is needed. Why was there not a total hysterectomy performed?

Lines 149-162:  A more complete medical history of the human cases should be provided, such as previous TOPs, and other invasive procedures prior to the 3 month period.

Line 165: There is a risk that an insemination catheter inserted through the cervix may have carried a component of the vaginal / cervical microbiome from the external cervical os into the uterus. Why was a double guarded system not used?

Line 321-322 and 329: Sentences needs to be reworked to make sense.

Lines 514-517: This sentence is difficult to understand.

Lines 533-537: the use of ‘should’ does not make sense.

Line 540: Define DEGs

Lines 595,7: ‘A’ should be in front of previous – A previous study

Line 613: Is the OTC used IV or intrauterine?

Line 698: displayed is not the word to use… indicated may be more appropriate.

Lines 741-747: this is very difficult to follow.

Author Response

Dear Reviewer,

Thank you very much for taking time out of your busy schedule to read this article. Thank for providing many constructive modification comments and suggestions, which have greatly helped us to further improve this manuscript. We carefully thought and analyzed your valuable opinions. On this basis, we have made the appropriate revisions and responses to your comments and suggestions, as noted below.

There are many grammatical / translational errors that need to be addressed:

Line 42: change 'unveiled' to 'found'

Response: “our findings unveiled” has been changed to “our results suggested”.

Line 45: the cause factor should be causal factors, or contributing factors.

Response: “cause factor” has been changed to “causal factors”.

Different fonts sizes, such as in lines 56-58 need to be corrected throughout the manuscript

Response: All the fonts sizes have been unified throughout the manuscript.

Lines 86-87: I think you need to put in (CE) following chronic endometritis

Response: Thanks. (CE) has been put in following chronic endometritis.

Lines 93-94: ‘have’ not ‘has’… and ‘of’ not ‘in’…  look for many similar errors

Response: Changes have been done.

Line 102: change pouch of Douglas to recto-genital pouch

Response: “Pouch of Douglas” has been changed to “Recto-Uterine Pouch”.

Lines 99-108: I suggest changing ‘canal’ to ‘lumen’

Response: The word comes from the original document. We politely asked to keep it as it was.

Lines 109-110: However, they do not have a duplex cervix. What are the risks of cross migration / contamination?

Response: Our experiment for the construction of IUA in rat model was totally carried out according to the commonly used protocol as described in a previous study (as ref 29 in our manuscript, Guo LP, Chen LM, Chen F, Jiang NH, Sui L. Smad signaling coincides with epithelial-mesenchymal transition in a rat model of intrauterine adhesion. Am J Transl Res. 2019 Aug 15;11(8):4726-4737. PMID: 31497194; PMCID: PMC6731410), in which, a mechanical injury was performed on the left uterus (IUA-L), and the right uterus (IUA-R, Sham-R) was used as the control. Because the non-IUA horn is connected to the IUA-horn, which does allow the IUA associated microbes to spread and to affect their environment, and some microbial species that prefer the scarred compartments in the same system could be observed in our study. Thus, the non-IUA horn shared the same microbe with IUA-horn due to the spread of the IUA associated microbes and their affection. Though the non-IUA horn shared the same microbial species with IUA, the distinct surgical treatments between two horns resulted in distinct growth rates and abundances of the microbes in these two different horns, which allowed us to distinguish the microbial causal factor. In order to avoid the possible false results, the association of the microbial factor with IUA was further elucidated by comprehensive analysis of the phenotypic characteristics, transcriptome and metabolic characteristics in the rat IUA and non-IUA models, and detailed comparison of the metabolic characteristics in human IUA and rat IUA.

Line 119: OTC has known tissue toxicity, this is a harsh product to use.

Response: We agree with the reviewer’s comment that OTC is a harsh product to use. However, we still found that it had been used for treatment of pathogen infection in some studies (please see refs 48, 53, 56).

Line 145: another common error that needs to be corrected throughout – ‘the’ needs to be inserted between operation on ‘the’ uteruses… actually uteri is the proper term. There are many such instances throughout the manuscript, I use this as one example.

Response: Thanks for the correction. All the “uteruses” have been changed to “uteri” throughout the manuscript.

Line 146: scrapped down is confusing. A more complete description of how the endometrium was obtained is needed. Why was there not a total hysterectomy performed?

Response: “scrapped” has been changed to “collected”. Our experiment for the construction of IUA in rat model was totally carried out according to the commonly used protocol as described in a previous study (as ref 29 in our manuscript, Guo LP, Chen LM, Chen F, Jiang NH, Sui L. Smad signaling coincides with epithelial-mesenchymal transition in a rat model of intrauterine adhesion. Am J Transl Res. 2019 Aug 15;11(8):4726-4737. PMID: 31497194; PMCID: PMC6731410). Thus, this reference for how the endometrium was obtained was cited in the manuscript. We also gave a description of how the IUA was performed as following: “The endometrial samples were collected with a mini-endometrial curette via scrapping off from inter-uterine.”

The rat horn is too tiny to perform a total hysterectomy analysis, so the phenotypic diagnostic characteristics were used for IUA identification as following: The myometria and endometria in non-IUA control group were clearly demarcated, and endometrial stromal cells were orderly distributed with normal size (Figs. 1A-1D, in blue circle). The glandular and endometrial epithelia were intact, and no obvious fibrosis was observed in non-IUA control group. Moreover, endometrial glands in IUA group decreased at 10.16±8.56 with 55% lower than those in non-IUA group at 4.58±4.56 (Fig. 1E). Micro-vessels (19.4±5.78) in IUA group were 24% higher than those in non-IUA control group (24.43±6.89) (Fig. 1F). Importantly, the fibrotic area in the IUA group (38.71±12.6) was 134.6% higher than that in non-IUA control group (16.5±7.75%) (Fig. 1G). About 40% TGF-β1 in endometrial epithelium was expressed in IUA group while no immunoreactivity for TGF-β1 expression in non-IUA group was observed (Fig. 1H). These results suggested that IUA in the rat model was successfully constructed, and matched diagnostic characteristics observed in human patients [13].

Lines 149-162:  A more complete medical history of the human cases should be provided, such as previous TOPs, and other invasive procedures prior to the 3 month period.

Response:   A more complete medical history of the human cases, including previous TOPs, and other invasive procedures prior to the 3 month period, has been provided in the supporting information.

Line 165: There is a risk that an insemination catheter inserted through the cervix may have carried a component of the vaginal / cervical microbiome from the external cervical os into the uterus. Why was a double guarded system not used?

Response: We agree with the reviewer’s comments that there is a risk that an insemination catheter inserted through the cervix may have carried a component of the vaginal / cervical microbiome from the external cervical os into the uterus. So we chosed the metaoblic profiles for analysis for the difference between human IUA and endometritis, because the potential contamination would make less effects on the metabolic profile than on the microbial data. We are very sorry that a double guarded system is temporarily unavailable in the hospital where we are carrying out the study. We believe that our finding that M. pulmonis is a key microbial factor for IUA, would help improve protection procedures against Mycoplasmopsis infection during the surgical performance on uterine when our study will be published. Our next investigation on IUA will focus on a specific protocol for inhibiting Mycoplasmopsis infection.

Line 321-322 and 329: Sentences needs to be reworked to make sense.

Response: The sentence in lines 321-322 has been changed to “IUA Rats in the control group were fed with sterile distilled water at a dose of 2 mL/d for 3 estrous cycles.” The sentence in line 329 has been changed to “One horn of the unterine of 46 rats of 8 weeks with average weight of 200g was established with IUA and the other horn without surgical operation was used as control”.

Lines 514-517: This sentence is difficult to understand.

Response: The sentence in Lines 514-517 has been changed to “To collect any endometrial samples from these patients’ uterine with any surgical tools was infeasible due to potential trauma injury.”

Lines 533-537: the use of ‘should’ does not make sense.

Response: “should be” has been changed to “was”.

Line 540: Define DEGs

Response: “differentially expressed genes” has been added before DEG.

Lines 595,7: ‘A’ should be in front of previous – A previous study

Response: “A” has been added.

Line 613: Is the OTC used IV or intrauterine?

Response: OTC was used via gavage for 21 days, which was described in the manuscript as following: Once IUA model caused by uterine scratch were observed, IUA rats were given these three medicine via gavage for 21 days and sterile distilled water was used as control group (about four estrous cycles).

Line 698: displayed is not the word to use… indicated may be more appropriate.

Response: “displayed” has been changed to “indicated”.

Lines 741-747: this is very difficult to follow.

Response: The long sentence in Lines 741-747 has been separated into two sentences as following: As expected, the two IUA highly enriched pathways before treatment, ABC transporters and quorum sensing that were both positively correlated with M. pulmonis (Fig. 3C), were significantly down-regulated in E2 group compared with IUA control after treatment (Fig. 8G). It was interesting to note that one IUA highly enriched pathway before treatment, linoleic acid metabolism (Figs. 4C-4D), was further significantly up-regulated in E2 group compared with IUA control after treatment.

Reviewer 3 Report

The manuscript by Ya et al. used a rat model to investigate the association between the uterine microbiome and Intrauterine Adhesion (IUA) formation. The hypothesis and in vivo model are sound; however, this reviewer is not convinced that the authors can test the hypothesis with the methodology used in the study. They are interested to link intrauterine microbiome to IUA. For this, they cause IUA in rats and search for changes in uterine microbiome, as well as endometrial metabolome and transcriptome, as a causal effect. However, differences may only be a consequence of the procedure/disease state rather than the cause. It would make more sense if authors reshaped their hypothesis and looked for changes in the uterine microbiome/metabolome/transcriptome as result to IUA vs non-IUA. In addition, the manuscript has numerous grammar and language issues, which need to be addressed. 

MAJOR:

Regarding the procedure to cause IUA, what was the sterility status of the surgical procedure? Were the needles used to scrape the endometria sterile? This reviewer wonders whether pathogens were introduced into the uterus during the procedure. 

Lines 147-148: “All the samples were histologically confirmed”. It is not clear to this reviewer what authors meant. Please clarify. 

Line 318: “0.003mg/d”. It is unclear what mg/d means. Please clarify. 

Line 335: What is the unit for “10.16±8.56”? The same applies for other measurements in the same paragraph. 

Lines 339-340: “About 40% TGF-β1 in endometrial epithelium was expressed in IUA group” is unclear, please rephrase. 

Regarding immunohistochemistry, authors report “proportion of stained cells”, however, it is unclear how they arrived at this number. Please clarify.

Regarding estimation of number of endometrial glands, it is unclear how the measurements were taken. Did authors count the number of glands per image field? Did they use any normalization factor? Was stage of estrous cycle taken into account?

Author should indicate in the statistical analyses section or in legends whether error bars represent SD or SEM. 

In Fig. 1, reviewers state that the yellow rectangle is positioned at “expanded endometrial glands at the secretion stage’, however, this reviewer believes that it is rather the uterine lumen. Please comment on this.  

In Fig. 2, it is clear that Control and non-IUA rats display contrasting microbial community. Authors should provide an explanation for such differences. These results make this reviewer wonder about introduction of pathogens into the uterus during the procedure to cause IUA. 

Fig. 4 panels E and F, what do the asterisks indicate?

Regarding the endometrial transcriptome data, did the authors correct the p-values for multiple comparisons? If not, an explanation for not doing so should be provided. 

Lines 322-323: Authors report that “The endometrial tissues were taken for further analysis after 3 estrous cycles (about 21 days)”; however, in lines 621-622 it is stated that “IUA rats were given these three medicine via gavage for 21 days and sterile distilled water was used as control group (about four estrous cycles)”. Authors should clarify on this. 

The TGF-β1 staining pattern in IUA samples are largely different between Fig. 1 and 6. Representative images of each group should be provided. 

Lines 680-682: “Most recent study reported that Staphylococcus spp. isolates from tonsils of slaughtered pigs presented high antibiotic resistance frequencies to OTC (97.1%) [56]. This was inconsistent with our results that OTC might not inhibit Staphylococcus microbiota in this study”. This statement leaves confusion. Authors should check for clarity. 

Lines 874-877: “We thus compared the metabolic profiles of uterine flushing fluids from IUA patients and endometritis patients and found out that two uterine flushing fluids were quite different, indicating that the endometritis-associated M. genitalium should not be the microbe for IUA formation in human”. This reviewer does not agree with this statement. Differences in uterine metabolomic profiles likely reflect the different disease states (endometritis vs IUA), therefore, one cannot arrive to that conclusion regarding M. genitalium and IUA. 

MINOR:

Lines 267-268: “To check the robustness and predictive ability of the OPLS-DA model, and a 200-time permutation testing was conducted” is not clear. Please reword. 

Line 295: “Using a Trizol-based method to extract the total RNA from rodent endometria” is not clear. Please reword. 

Line 321: “in control the group” should be “in the control group”.

Fig. 4, in panel (A), the brackets denotating IUA and non-IUA groups are out of place. Please correct. 

Lines 511-512: “Considering that M. genitalium, colonized in human genital tract, played an important role in pelvic inflammatory disease and acute endometritis” is not clear. Please reword. 

Lines 523-526: This reviewer does not agree with the statement that “Obviously, the number (10) of the differentially regulated metabolites in EM vs FTO were far less than those of human IUA vs FTO or EM, indicating that IUA in human could cause much more metabolic alteration than endometritis and FTO”. To be able to state this, the authors should had analyzed a cohort of samples collected from non-diseased uteri. 

Line 598: “endometrial” should be “endometrium”.

Author Response

Dear Reviewer,

Thank you very much for taking time out of your busy schedule to read this article. Thank for providing many constructive modification comments and suggestions, which have greatly helped us to further improve this manuscript. We carefully thought and analyzed your valuable opinions. On this basis, we have made the appropriate revisions and responses to your comments and suggestions, as noted below.

The manuscript by Ya et al. used a rat model to investigate the association between the uterine microbiome and Intrauterine Adhesion (IUA) formation. The hypothesis and in vivo model are sound; however, this reviewer is not convinced that the authors can test the hypothesis with the methodology used in the study. They are interested to link intrauterine microbiome to IUA. For this, they cause IUA in rats and search for changes in uterine microbiome, as well as endometrial metabolome and transcriptome, as a causal effect. However, differences may only be a consequence of the procedure/disease state rather than the cause. It would make more sense if authors reshaped their hypothesis and looked for changes in the uterine microbiome/metabolome/transcriptome as result to IUA vs non-IUA. In addition, the manuscript has numerous grammar and language issues, which need to be addressed. 

MAJOR:

 Regarding the procedure to cause IUA, what was the sterility status of the surgical procedure? Were the needles used to scrape the endometria sterile? This reviewer wonders whether pathogens were introduced into the uterus during the procedure. 

 Response: Our experiment for the construction of IUA in rat model was carried out according to the commonly used protocol as described in a previous study [29], which involved the protection procedures against microbial infection during the surgical performance on uterine. The sterility status of the surgical procedure meets the medical requirements. All the surgical tools including the needles used to scrape the endometria were sterile. Moreover, 46 rats that were divided into 3 batches were used for the IUA models in our study. In this study, what we could do was to find out a key microbial factor for IUA. We first evaluated the uterine horns with the diagnostic characteristics for IUA. Then the IUA samples were analyzed for the microbiome taxonomy, genomic, transcriptome and metabolic characteristics. The distinct growth rates and abundances of the microbes between IUA and non-IUA allowed us to distinguish the microbial factor. In order to avoid the possible false results, the association of the microbial factor with IUA endometrium was further elucidated by comprehensive analysis of endometrial transcriptome and metabolic characteristics in the rat IUA and non-IUA models, and detailed comparison of the metabolic characteristics in human IUA and rat IUA. Our finding that M. pulmonis is a key microbial factor for IUA, would help improve protection against Mycoplasmopsis infection during the surgical performance on uterine when our study will be published. Our next investigation on IUA will focus on a specific protocol for inhibiting Mycoplasmopsis infection.

Lines 147-148: “All the samples were histologically confirmed”. It is not clear to this reviewer what authors meant. Please clarify. 

 Response: This sentence has been changed to “All the samples were histologically evaluated for the diagnostic characteristics for IUA ”.

Line 318: “0.003mg/d”. It is unclear what mg/d means. Please clarify. 

 Response: This has been changed to 0.003 mg per day.

Line 335: What is the unit for “10.16±8.56”? The same applies for other measurements in the same paragraph. 

 Response: The value10.16±8.56 indicated the number of glands, as described in the reference (Li B, Zhang Q, Sun J, Lai D. Human amniotic epithelial cells improve fertility in an intrauterine adhesion mouse model. Stem Cell Res Ther. 2019;10(1):257. Published 2019 Aug 14. doi:10.1186/s13287-019-1368-9), as following:

Lines 339-340: “About 40% TGF-β1 in endometrial epithelium was expressed in IUA group” is unclear, please rephrase. 

 Response: This sentence has been changed to “The immunoreactivity level for TGF-β1 expression in endometrial epithelium of IUA group was about 40%”.

Regarding immunohistochemistry, authors report “proportion of stained cells”, however, it is unclear how they arrived at this number. Please clarify.

Response: According to a previous study (Zhang J, Li Z, Chen F, et al. TGF-β1 suppresses CCL3/4 expression through the ERK signaling pathway and inhibits intervertebral disc degeneration and inflammation-related pain in a rat model. Exp Mol Med. 2017;49(9):e379. Published 2017 Sep 22. doi:10.1038/emm.2017.136), the immunopositive cells in the endometrium were counted in 10 high-power fields (x 400) by independent researchers blinded to information pertaining to the study, and the percentage of immunopositive cells was calculated by dividing the number of immunopositive cells by the total number of cells and then multiplying the resulting number by 100. This has been clarified in the manuscript.

Regarding estimation of number of endometrial glands, it is unclear how the measurements were taken. Did authors count the number of glands per image field? Did they use any normalization factor? Was stage of estrous cycle taken into account?

Response: Please see the above response. Yes, we counted the number of glands per image field. Because under the same high-power fields (x 400), we didn’t use any normalization factor. The stage of estrous cycle has been taken into account. This has been clarified in the manuscript.

Author should indicate in the statistical analyses section or in legends whether error bars represent SD or SEM. 

Response: Error bars represent SD. When comparing the differences of a component between groups, mean was used to represent the value of that in the group, and standard deviation (SD) were shown in the figure as error bars. Microbial community differences analysis between groups were computed using permutational multivariate analysis of variance (PERMANOVA). This has been clarified in the manuscript.

In Fig. 1, reviewers state that the yellow rectangle is positioned at “expanded endometrial glands at the secretion stage’, however, this reviewer believes that it is rather the uterine lumen. Please comment on this.

Response: Our results were consistent with the reference (Li B, Zhang Q, Sun J, Lai D. Human amniotic epithelial cells improve fertility in an intrauterine adhesion mouse model. Stem Cell Res Ther. 2019;10(1):257. Published 2019 Aug 14. doi:10.1186/s13287-019-1368-9), that control had a large edndometrial gland (the center in A as following). Thus, the yellow rectangle in our study referred to the similar expanded endometrial glands at the secretion stage’.

In Fig. 2, it is clear that Control and non-IUA rats display contrasting microbial community. Authors should provide an explanation for such differences. These results make this reviewer wonder about introduction of pathogens into the uterus during the procedure to cause IUA. 

Response: Because the non-IUA horn is connected to the IUA-horn, which does allow the IUA associated microbes to spread and to affect their environment, and some microbial species that prefer the scarred compartments in the same system could be observed in our study. Thus, the non-IUA horn shared the same microbe with IUA-horn due to the spread of the IUA associated microbes and their affection. Though the non-IUA horn shared the same microbial species with IUA, the distinct surgical treatments between two horns resulted in distinct growth rates and abundances of the microbes in these two different horns, which allowed us to distinguish the microbial factor. In this study, what we could do was to find out a key microbial factor associated with IUA. We believe that our finding that M. pulmonis is a key microbial factor for IUA, would help improve protection procedures against Mycoplasmopsis infection during the surgical performance on uterine when our study will be published. Our next investigation on IUA will focus on a specific protocol for inhibiting Mycoplasmopsis infection.

Fig. 4 panels E and F, what do the asterisks indicate?

Response:  * refers to P < 0.05. This has been clarified in Figure 4.

Regarding the endometrial transcriptome data, did the authors correct the p-values for multiple comparisons? If not, an explanation for not doing so should be provided. 

Response: Benjamini-Hochberg (BH) method was used to adjust p-value computed from corresponding significant differences test. This has been clarified in the methods.

Lines 322-323: Authors report that “The endometrial tissues were taken for further analysis after 3 estrous cycles (about 21 days)”; however, in lines 621-622 it is stated that “IUA rats were given these three medicine via gavage for 21 days and sterile distilled water was used as control group (about four estrous cycles)”. Authors should clarify on this. 

Response: Seventy-two rats were used in the full study. Among them, 49 rats (group one) were used for characterization of the microbial and endometrial factors involved in IUA formation. In 46 rats (group one), standard scratch method for IUA formation was applied in one uterine horn (total 46 horns) and the other uterine horn without scratch served as a non-IUA control (total 46 horns). The uterine horns from three rats without surgery were used as control (total 6 horns). Adhesion scores were compared 3 weeks later and 16S rRNA analysis, deep metagenomic sequencing, metabolomics pro­filing and endometrial transcriptional analysis of IUA and Non-IUA groups were performed. In the second group, 23 animals were operated on. Both uterine horns (total 36 horns) of these 23 rats were inflicted with IUA formation. After 3 weeks, these rats were treated with Estrogen (n=5), Oxytetracycline (n=6) and Gongxuening (n=6), respectively, through oral administration every day for 3 weeks. Six rats treated with water served as a IUA control. Adhesion scores were compared 3 weeks later and 16S rRNA analysis and endometrial transcriptional analysis of IUA and Non-IUA groups were performed. The above clarification has been also provided in the manuscript.

The TGF-β1 staining pattern in IUA samples are largely different between Fig. 1 and 6. Representative images of each group should be provided. 

Response: Correction has been done.

Lines 680-682: “Most recent study reported that Staphylococcus spp. isolates from tonsils of slaughtered pigs presented high antibiotic resistance frequencies to OTC (97.1%) [56]. This was inconsistent with our results that OTC might not inhibit Staphylococcus microbiota in this study”. This statement leaves confusion. Authors should check for clarity. 

 Response:

Response: Thanks for the suggestion. Here are a typo. “inconsistent” has been changed to “consistent”.

Lines 874-877: “We thus compared the metabolic profiles of uterine flushing fluids from IUA patients and endometritis patients and found out that two uterine flushing fluids were quite different, indicating that the endometritis-associated M. genitalium should not be the microbe for IUA formation in human”. This reviewer does not agree with this statement. Differences in uterine metabolomic profiles likely reflect the different disease states (endometritis vs IUA), therefore, one cannot arrive to that conclusion regarding M. genitalium and IUA. 

Response: To better clarify it, this sentence has been changed to “indicating that there might be distinct associated microbial factors between endometritis and IUA”. 

MINOR:

 Lines 267-268: “To check the robustness and predictive ability of the OPLS-DA model, and a 200-time permutation testing was conducted” is not clear. Please reword. 

Response: This sentence has been changed to “Supervised PLS-DA was utilized to analysis group separation of metabolomics data and mine the variables responsible for classification. The robustness and predictive ability of the estimation model were verified by 7-fold cross-validation, and the model was further validated by permutation test with 200 permutations.”

Line 295: “Using a Trizol-based method to extract the total RNA from rodent endometria” is not clear. Please reword. 

Response: This has been changed to “A Trizol-based method was applied to extract the total RNA from rodent endometrial.”

Line 321: “in control the group” should be “in the control group”.

Response: Correction has been done.

Fig. 4, in panel (A), the brackets denotating IUA and non-IUA groups are out of place. Please correct. 

 Response: Correction has been done.

Lines 511-512: “Considering that M. genitalium, colonized in human genital tract, played an important role in pelvic inflammatory disease and acute endometritis” is not clear. Please reword. 

 Response: This sentence has been changed to “Previous studies reported that M. genitalium played an important role in inflammatory disease and acute endometritis”.

Lines 523-526: This reviewer does not agree with the statement that “Obviously, the number (10) of the differentially regulated metabolites in EM vs FTO were far less than those of human IUA vs FTO or EM, indicating that IUA in human could cause much more metabolic alteration than endometritis and FTO”. To be able to state this, the authors should had analyzed a cohort of samples collected from non-diseased uteri. 

 Response: Patients with blocked fallopian tube were used as non-IUA control due to two reasons. 1) Patients with blocked fallopian tube have an intact and healthy uterus, namely non-diseased uteri; 2) The flushing fluids from non-diseased genitals could not be accessible due to lack of healthy volunteers.

Line 598: “endometrial” should be “endometrium”.

Response: Change has been done.

Dear Reviewer,

Thank you very much for taking time out of your busy schedule to read this article. Thank for providing many constructive modification comments and suggestions, which have greatly helped us to further improve this manuscript. We carefully thought and analyzed your valuable opinions. On this basis, we have made the appropriate revisions and responses to your comments and suggestions, as noted below.

The manuscript by Ya et al. used a rat model to investigate the association between the uterine microbiome and Intrauterine Adhesion (IUA) formation. The hypothesis and in vivo model are sound; however, this reviewer is not convinced that the authors can test the hypothesis with the methodology used in the study. They are interested to link intrauterine microbiome to IUA. For this, they cause IUA in rats and search for changes in uterine microbiome, as well as endometrial metabolome and transcriptome, as a causal effect. However, differences may only be a consequence of the procedure/disease state rather than the cause. It would make more sense if authors reshaped their hypothesis and looked for changes in the uterine microbiome/metabolome/transcriptome as result to IUA vs non-IUA. In addition, the manuscript has numerous grammar and language issues, which need to be addressed. 

MAJOR:

 Regarding the procedure to cause IUA, what was the sterility status of the surgical procedure? Were the needles used to scrape the endometria sterile? This reviewer wonders whether pathogens were introduced into the uterus during the procedure. 

 Response: Our experiment for the construction of IUA in rat model was carried out according to the commonly used protocol as described in a previous study [29], which involved the protection procedures against microbial infection during the surgical performance on uterine. The sterility status of the surgical procedure meets the medical requirements. All the surgical tools including the needles used to scrape the endometria were sterile. Moreover, 46 rats that were divided into 3 batches were used for the IUA models in our study. In this study, what we could do was to find out a key microbial factor for IUA. We first evaluated the uterine horns with the diagnostic characteristics for IUA. Then the IUA samples were analyzed for the microbiome taxonomy, genomic, transcriptome and metabolic characteristics. The distinct growth rates and abundances of the microbes between IUA and non-IUA allowed us to distinguish the microbial factor. In order to avoid the possible false results, the association of the microbial factor with IUA endometrium was further elucidated by comprehensive analysis of endometrial transcriptome and metabolic characteristics in the rat IUA and non-IUA models, and detailed comparison of the metabolic characteristics in human IUA and rat IUA. Our finding that M. pulmonis is a key microbial factor for IUA, would help improve protection against Mycoplasmopsis infection during the surgical performance on uterine when our study will be published. Our next investigation on IUA will focus on a specific protocol for inhibiting Mycoplasmopsis infection.

Lines 147-148: “All the samples were histologically confirmed”. It is not clear to this reviewer what authors meant. Please clarify. 

 Response: This sentence has been changed to “All the samples were histologically evaluated for the diagnostic characteristics for IUA ”.

Line 318: “0.003mg/d”. It is unclear what mg/d means. Please clarify. 

 Response: This has been changed to 0.003 mg per day.

Line 335: What is the unit for “10.16±8.56”? The same applies for other measurements in the same paragraph. 

 Response: The value10.16±8.56 indicated the number of glands, as described in the reference (Li B, Zhang Q, Sun J, Lai D. Human amniotic epithelial cells improve fertility in an intrauterine adhesion mouse model. Stem Cell Res Ther. 2019;10(1):257. Published 2019 Aug 14. doi:10.1186/s13287-019-1368-9), as following:

Lines 339-340: “About 40% TGF-β1 in endometrial epithelium was expressed in IUA group” is unclear, please rephrase. 

 Response: This sentence has been changed to “The immunoreactivity level for TGF-β1 expression in endometrial epithelium of IUA group was about 40%”.

Regarding immunohistochemistry, authors report “proportion of stained cells”, however, it is unclear how they arrived at this number. Please clarify.

Response: According to a previous study (Zhang J, Li Z, Chen F, et al. TGF-β1 suppresses CCL3/4 expression through the ERK signaling pathway and inhibits intervertebral disc degeneration and inflammation-related pain in a rat model. Exp Mol Med. 2017;49(9):e379. Published 2017 Sep 22. doi:10.1038/emm.2017.136), the immunopositive cells in the endometrium were counted in 10 high-power fields (x 400) by independent researchers blinded to information pertaining to the study, and the percentage of immunopositive cells was calculated by dividing the number of immunopositive cells by the total number of cells and then multiplying the resulting number by 100. This has been clarified in the manuscript.

Regarding estimation of number of endometrial glands, it is unclear how the measurements were taken. Did authors count the number of glands per image field? Did they use any normalization factor? Was stage of estrous cycle taken into account?

Response: Please see the above response. Yes, we counted the number of glands per image field. Because under the same high-power fields (x 400), we didn’t use any normalization factor. The stage of estrous cycle has been taken into account. This has been clarified in the manuscript.

Author should indicate in the statistical analyses section or in legends whether error bars represent SD or SEM. 

Response: Error bars represent SD. When comparing the differences of a component between groups, mean was used to represent the value of that in the group, and standard deviation (SD) were shown in the figure as error bars. Microbial community differences analysis between groups were computed using permutational multivariate analysis of variance (PERMANOVA). This has been clarified in the manuscript.

In Fig. 1, reviewers state that the yellow rectangle is positioned at “expanded endometrial glands at the secretion stage’, however, this reviewer believes that it is rather the uterine lumen. Please comment on this.

Response: Our results were consistent with the reference (Li B, Zhang Q, Sun J, Lai D. Human amniotic epithelial cells improve fertility in an intrauterine adhesion mouse model. Stem Cell Res Ther. 2019;10(1):257. Published 2019 Aug 14. doi:10.1186/s13287-019-1368-9), that control had a large edndometrial gland (the center in A as following). Thus, the yellow rectangle in our study referred to the similar expanded endometrial glands at the secretion stage’.

In Fig. 2, it is clear that Control and non-IUA rats display contrasting microbial community. Authors should provide an explanation for such differences. These results make this reviewer wonder about introduction of pathogens into the uterus during the procedure to cause IUA. 

Response: Because the non-IUA horn is connected to the IUA-horn, which does allow the IUA associated microbes to spread and to affect their environment, and some microbial species that prefer the scarred compartments in the same system could be observed in our study. Thus, the non-IUA horn shared the same microbe with IUA-horn due to the spread of the IUA associated microbes and their affection. Though the non-IUA horn shared the same microbial species with IUA, the distinct surgical treatments between two horns resulted in distinct growth rates and abundances of the microbes in these two different horns, which allowed us to distinguish the microbial factor. In this study, what we could do was to find out a key microbial factor associated with IUA. We believe that our finding that M. pulmonis is a key microbial factor for IUA, would help improve protection procedures against Mycoplasmopsis infection during the surgical performance on uterine when our study will be published. Our next investigation on IUA will focus on a specific protocol for inhibiting Mycoplasmopsis infection.

Fig. 4 panels E and F, what do the asterisks indicate?

Response:  * refers to P < 0.05. This has been clarified in Figure 4.

Regarding the endometrial transcriptome data, did the authors correct the p-values for multiple comparisons? If not, an explanation for not doing so should be provided. 

Response: Benjamini-Hochberg (BH) method was used to adjust p-value computed from corresponding significant differences test. This has been clarified in the methods.

Lines 322-323: Authors report that “The endometrial tissues were taken for further analysis after 3 estrous cycles (about 21 days)”; however, in lines 621-622 it is stated that “IUA rats were given these three medicine via gavage for 21 days and sterile distilled water was used as control group (about four estrous cycles)”. Authors should clarify on this. 

Response: Seventy-two rats were used in the full study. Among them, 49 rats (group one) were used for characterization of the microbial and endometrial factors involved in IUA formation. In 46 rats (group one), standard scratch method for IUA formation was applied in one uterine horn (total 46 horns) and the other uterine horn without scratch served as a non-IUA control (total 46 horns). The uterine horns from three rats without surgery were used as control (total 6 horns). Adhesion scores were compared 3 weeks later and 16S rRNA analysis, deep metagenomic sequencing, metabolomics pro­filing and endometrial transcriptional analysis of IUA and Non-IUA groups were performed. In the second group, 23 animals were operated on. Both uterine horns (total 36 horns) of these 23 rats were inflicted with IUA formation. After 3 weeks, these rats were treated with Estrogen (n=5), Oxytetracycline (n=6) and Gongxuening (n=6), respectively, through oral administration every day for 3 weeks. Six rats treated with water served as a IUA control. Adhesion scores were compared 3 weeks later and 16S rRNA analysis and endometrial transcriptional analysis of IUA and Non-IUA groups were performed. The above clarification has been also provided in the manuscript.

The TGF-β1 staining pattern in IUA samples are largely different between Fig. 1 and 6. Representative images of each group should be provided. 

Response: Correction has been done.

Lines 680-682: “Most recent study reported that Staphylococcus spp. isolates from tonsils of slaughtered pigs presented high antibiotic resistance frequencies to OTC (97.1%) [56]. This was inconsistent with our results that OTC might not inhibit Staphylococcus microbiota in this study”. This statement leaves confusion. Authors should check for clarity. 

 Response:

Response: Thanks for the suggestion. Here are a typo. “inconsistent” has been changed to “consistent”.

Lines 874-877: “We thus compared the metabolic profiles of uterine flushing fluids from IUA patients and endometritis patients and found out that two uterine flushing fluids were quite different, indicating that the endometritis-associated M. genitalium should not be the microbe for IUA formation in human”. This reviewer does not agree with this statement. Differences in uterine metabolomic profiles likely reflect the different disease states (endometritis vs IUA), therefore, one cannot arrive to that conclusion regarding M. genitalium and IUA. 

Response: To better clarify it, this sentence has been changed to “indicating that there might be distinct associated microbial factors between endometritis and IUA”. 

MINOR:

 Lines 267-268: “To check the robustness and predictive ability of the OPLS-DA model, and a 200-time permutation testing was conducted” is not clear. Please reword. 

Response: This sentence has been changed to “Supervised PLS-DA was utilized to analysis group separation of metabolomics data and mine the variables responsible for classification. The robustness and predictive ability of the estimation model were verified by 7-fold cross-validation, and the model was further validated by permutation test with 200 permutations.”

Line 295: “Using a Trizol-based method to extract the total RNA from rodent endometria” is not clear. Please reword. 

Response: This has been changed to “A Trizol-based method was applied to extract the total RNA from rodent endometrial.”

Line 321: “in control the group” should be “in the control group”.

Response: Correction has been done.

Fig. 4, in panel (A), the brackets denotating IUA and non-IUA groups are out of place. Please correct. 

 Response: Correction has been done.

Lines 511-512: “Considering that M. genitalium, colonized in human genital tract, played an important role in pelvic inflammatory disease and acute endometritis” is not clear. Please reword. 

 Response: This sentence has been changed to “Previous studies reported that M. genitalium played an important role in inflammatory disease and acute endometritis”.

Lines 523-526: This reviewer does not agree with the statement that “Obviously, the number (10) of the differentially regulated metabolites in EM vs FTO were far less than those of human IUA vs FTO or EM, indicating that IUA in human could cause much more metabolic alteration than endometritis and FTO”. To be able to state this, the authors should had analyzed a cohort of samples collected from non-diseased uteri. 

 Response: Patients with blocked fallopian tube were used as non-IUA control due to two reasons. 1) Patients with blocked fallopian tube have an intact and healthy uterus, namely non-diseased uteri; 2) The flushing fluids from non-diseased genitals could not be accessible due to lack of healthy volunteers.

Line 598: “endometrial” should be “endometrium”.

Response: Change has been done.

Round 2

Reviewer 1 Report

Dear Authors,

Thank you for the answers and the corrections. Some confusion with the taxonomy is however still present. As far as I can judge, the current nomenclature is Mycoplasmopsis pulmonis for rat pathogen and Mycoplasma pneumoniae for human one (and Mycoplasma genitalium in referenced source).

There are still major questions on how the role of the Mycoplasma / Mycoplasmopsis is introduced in this study. The role of bacteria in human IUA is a valuable hypothesis. However, one can use the experimental data to draw the immediate conclusions only. And then, one may speculate on the basis of the current and the referenced studies, and to introduce the hypothesis.

 1. In the current study, there is a limitation of lacking some sorts of controls. It is understandable that the control groups can not be introduced in this stage. Yet it is important to describe the limitations in the Discussion.

This study has found the association of the bacteria with IUA lesions, yet it should be better explored to rule out the association with the puncture of the uterus itself, and to compare the IUA state with completely ‘healthy’ one. The study that introduced the IUA model had the sham operation. The authors of the current study assumed the sham-operated uterus ought to be similar enough to the uterus in the intact rat. Yet the infection is a concern that arises anew during each set of experiments. Also, it was found during the current study that the healthy uterus may differ in microbiome from the non-IUA horns that were connected to IUA-model horns, making non-operated uterus a desirable control. These details were addressed by the authors in their Response, yet the authors are asked to address them in the Discussion as well. The limitations should be clearly stated.

 2. Chapter 3.5 suffers from the hypothesis being not clearly separated from data-driven conclusions. The circumstances only allowed for the analysis of the metabolites in human uteri. This means one can only draw data-supported conclusions on the similarities and differences in metabolites. The bacteria associated with endometritis were known from the literature (line 610–611). As the experiments in the current study did not look for bacteria in human IUA, they can not be used to answer the question “if there were any differences between the IUA-associated Mycoplasmopsis species and the endometritis-associated Mycoplasmopsis species in human” (lines 611–613) and to draw the conclusions about “IUA-associated Mycoplasmopsis species”. So far as there was no direct demonstration of Mycoplasma / Mycoplasmopsis in human IUA, only assumptions about it can be made. It is understandable that the additional sample analysis can not be carried out for the same patients. Still, the conclusions of the chapter can be rewritten (lines 630–636). The study of metabolites in humans, and of metabolites and bacteria in rats may be described as being suggestive of the possible role of the Mycoplasma involvement in human IUA. This is best if presented in the Discussion.

3. The authors’ explanation about their arrival to the possible role of M. pneumoniae is really interesting. However, the wording “it was most likely that human lung pathogen” that was involved (line 1004) is too strong. M. pneumoniae could be involved, was proposed to be involved, yet the evidence so far was not provided by the bacteriological analysis, so the use of the words for an assumption should be careful.

Author Response

Dear Reviewer,

Thank you very much for taking time out of your busy schedule to patiently read our responses. Thank for providing the constructive modification comments and suggestions, which have greatly helped us to further improve this manuscript. We carefully thought and analyzed your valuable opinions. On this basis, we have made the appropriate revisions and responses to your comments and suggestions, as noted below.

Some confusion with the taxonomy is however still present. As far as I can judge, the current nomenclature is Mycoplasmopsis pulmonis for rat pathogen and Mycoplasma pneumoniae for human one (and Mycoplasma genitalium in referenced source). There are still major questions on how the role of the Mycoplasma / Mycoplasmopsis is introduced in this study. The role of bacteria in human IUA is a valuable hypothesis. However, one can use the experimental data to draw the immediate conclusions only. And then, one may speculate on the basis of the current and the referenced studies, and to introduce the hypothesis.

  1. In the current study, there is a limitation of lacking some sorts of controls. It is understandable that the control groups can not be introduced in this stage. Yet it is important to describe the limitations in the Discussion.

This study has found the association of the bacteria with IUA lesions, yet it should be better explored to rule out the association with the puncture of the uterus itself, and to compare the IUA state with completely ‘healthy’ one. The study that introduced the IUA model had the sham operation. The authors of the current study assumed the sham-operated uterus ought to be similar enough to the uterus in the intact rat. Yet the infection is a concern that arises anew during each set of experiments. Also, it was found during the current study that the healthy uterus may differ in microbiome from the non-IUA horns that were connected to IUA-model horns, making non-operated uterus a desirable control. These details were addressed by the authors in their Response, yet the authors are asked to address them in the Discussion as well. The limitations should be clearly stated.

Response: Thanks for the suggestion. These details that were addressed by us in our last response has been added in the Discussion as following:

The association of multiple trauma and Mycoplasmopsis infection is well-known. It has been assumed that the trauma might give rise to Mycoplasmopsis spp. Our results that M. pulmonis infection occurred in the traumatized endometrium was consistent with the specific association. The non-IUA horn shared the same dominant M. pulmonis with IUA-horn most likely because the non-IUA horn was connected to the IUA-horn, allowing M. pulmonis to spread and to affect the environment, which could also explain why M. pulmonis was not detected in the intact and healthy uteri. However, the distinct growth rate and abundance of M. pulmonis between non-IUA horns and IUA horns caused by the trauma could allow us to distinguish M. pulmonis as the key microbial factor for IUA.

Though M. pulmonis predominated both the uterine horns in non-IUA and IUA groups, the non-IUA horns without the trauma to the endometrium was still healthy without IUA while the horn with the trauma to the endometrium formed IUA, consistent with previous study that the trauma injury to the endometrium was an indispensable factor for IUA. Previous study that the mechanical injury on the uteri without the damage to the endometrium, as a sham control, could not induce IUA and shared almost the same transcription of all the target genes with the other intact and healthy uteri controls. However, if the sham control could induce the M. pulmonis infection still remained unknown.  

  1. Chapter 3.5 suffers from the hypothesis being not clearly separated from data-driven conclusions. The circumstances only allowed for the analysis of the metabolites in human uteri. This means one can only draw data-supported conclusions on the similarities and differences in metabolites. The bacteria associated with endometritis were known from the literature (line 610–611). As the experiments in the current study did not look for bacteria in human IUA, they can not be used to answer the question “if there were any differences between the IUA-associated Mycoplasmopsisspecies and the endometritis-associated Mycoplasmopsis species in human” (lines 611–613) and to draw the conclusions about “IUA-associated Mycoplasmopsis species”. So far as there was no direct demonstration of Mycoplasma / Mycoplasmopsis in human IUA, only assumptions about it can be made. It is understandable that the additional sample analysis can not be carried out for the same patients. Still, the conclusions of the chapter can be rewritten (lines 630–636). The study of metabolites in humans, and of metabolites and bacteria in rats may be described as being suggestive of the possible role of the Mycoplasma involvement in human IUA. This is best if presented in the Discussion.

Response: Thanks for the nice suggestion. The conclusions in lines 630–636 have been rewritten as the following: “The similar IUA-associated metabolites between human and rat suggested the possible role of Mycoplasmopsis involvement in human IUA. The distinct metabolic alteration between human IUA and human endometritis indicated that the endometritis-associated M. genitalium in human might not be associated with IUA.” This similar clarification have been also stated in the Discussion.

  1. The authors’ explanation about their arrival to the possible role of M. pneumoniaeis really interesting. However, the wording “it was most likely that human lung pathogen” that was involved (line 1004) is too strong. M. pneumoniaecould be involved, was proposed to be involved, yet the evidence so far was not provided by the bacteriological analysis, so the use of the words for an assumption should be careful.

Response: Thanks for the suggestion. The sentence “it was most likely that human lung pathogen” in lines 1004-1006 has been deleted.

Reviewer 2 Report

The manuscript has been improved with the latest edits. However, the English translations is far from being correct. The grammar, sentence structure and word usage is inadequate for acceptance for publication. An example is the use of uterine in many instances where uterus should be used.  I am presuming the intended meaning of many sentences, by guessing which words the authors are intending, but not using.  I started to make editing corrections, but that is not the duty of a reviewer, it is the authors and then the editors.  Therefore, there must be major correction in this area before this manuscript is publishable in my opinion.  

Author Response

Dear Reviewer,

Thank you very much for taking time out of your busy schedule to patiently read our responses. Thank for providing the constructive modification comments and suggestions, which have greatly helped us to further improve this manuscript. We carefully thought and analyzed your valuable opinions. On this basis, we have made the appropriate revisions and responses to your comments and suggestions, as noted below.

Response:  The use of uterine in many instances where uterus should be used has been corrected. The paper has been edited by Professor Joan Bennett from Rutgers University. This has been clarified in the manuscript.

Reviewer 3 Report

All comments were addressed. 

Author Response

Many thanks for your great efforts!

Round 3

Reviewer 1 Report

Dear Authors, thank you for taking into account the suggestions.

Author Response

Dear Reviewer,

Thanks for your much valuable suggestions and great effort.

Reviewer 2 Report

There has been notable improvement. The manuscript is close to being reading for acceptance and much improved.

There remains several sentences still in need of editing. For example the sentence from line 103 - 105 is quite cumbersome. Others that I noted that need editing are 696-698; 700-703; 763-764; 777-778; 779; 888-890. See sentence starting on 977 as a good example.

Also, is the sentence 600-603 belong in the results section or in the discussion?

Author Response

Dear Reviewer,

Thank you very much for taking time out of your busy schedule to patiently read our responses. Thanks for pointing out the improper sentences. We carefully thought and analyzed these sentences. On this basis, we have made the appropriate revisions as noted below.

There remains several sentences still in need of editing. For example the sentence from line 103 - 105 is quite cumbersome. Others that I noted that need editing are 696-698; 700-703; 763-764; 777-778; 779; 888-890. See sentence starting on 977 as a good example.

Response: The sentence from line 103 – 105 has been changed to “Hormone estrogen has always been used to prevent postoperative re-adhesions because it can promote endometrium growth”.

The sentence from line 696698 has been changed to “A previous study reported that Mycoplasmopsis infection was associated with reproductive failure in a number of mammals, and administration of OTC to infertile couples could result in about 30% pregnancy increase”.

The sentence from line 700703 has been changed to “Thus, we examined the effects of these three medicines on IUA treatment after the IUA rats were given these medicines by gavage for 21 days (about four estrous cycles), control group was given sterile distilled water”.

The sentence from line 763-764 has been changed to “ A recent study reported that 97.1% of Staphylococcus spp. isolates from tonsils of slaughtered pigs displayed OTC resistance”.

The sentence from line 777-778 has been changed to “This result was consistent with the above result that Staphylococcus aureus infection was highly enriched KEGG pathway in IUA and a previous study that Staphylococcus microorganism could acutely exacerbate pulmonary fibrosis [46]”.

The sentence from line 779 has been changed to “The recent researches on Lactobacillus have been contradictory”.

The sentence from line 888-890 has been changed to “A previous study indicated that IUA patients displayed significantly higher IL-6 level in serum than non-IUA patients”.

Also, is the sentence 600-603 belong in the results section or in the discussion?

Response: We put the sentence in lines 600-603 in the result section 3.5 in order to bring out the crucial point for this section (Similar metabolites between human IUA and rat IUA while distinct metabolic profiles between human IUA and human endometritis). This was also in discussions with another sentences: We thus compared the metabolic profiles of uterine flushing fluids from IUA patients and endometritis patients and found out the possible role of Mycoplasmopsis involvement in human IUA based on the similar IUA-associated metabolites between humans and rats. Moreover, the distinct metabolic profiles of two uterine flushing fluids between human IUA and endometritis, indicated that the endometritis-associated M. genitalium should not be the microbe for IUA formation in humans.

Many thanks for your valuable suggestions and  the constructive modification comments,  which have greatly helped us to further improve this manuscript.

Best wishes!

Xuemei